# SESaMo: Symmetry-Enforcing Stochastic Modulation for Normalizing Flows

**Janik Kreit**[1,2*]   **Dominic Schuh**[1,2*]   **Kim A. Nicoli**[1,2,3*]   **Lena Funcke**[1,2*]

[1]Transdisciplinary Research Area (TRA) Matter, University of Bonn, Germany
[2]Helmholtz Institute for Radiation and Nuclear Physics (HISKP), University of Bonn, Germany
[3]Oldendorff Carriers GmbH & Co. KG, Lübeck, Germany

## Abstract

Deep generative models have recently garnered significant attention across various fields, from physics to chemistry, where sampling from unnormalized Boltzmann-like distributions represents a fundamental challenge. In particular, autoregressive models and normalizing flows have become prominent due to their appealing ability to yield closed-form probability densities. Moreover, it is well-established that incorporating prior knowledge—such as symmetries—into deep neural networks can substantially improve training performances. In this context, recent advances have focused on developing symmetry-equivariant generative models, achieving remarkable results. Building upon these foundations, this paper introduces Symmetry-Enforcing Stochastic Modulation (SESaMo). Similar to equivariant normalizing flows, SESaMo enables the incorporation of inductive biases (e.g., symmetries) into normalizing flows through a novel technique called *stochastic modulation*. This approach enhances the flexibility of the generative model by enforcing exact symmetries while, for the first time, enabling the model to learn broken symmetries during training. Our numerical experiments benchmark SESaMo in different scenarios, including an 8-Gaussian mixture model and physically relevant field theories, such as the $\phi^4$ theory and the Hubbard model.

## 1 Introduction

Sampling from unnormalized Boltzmann distributions is an ubiquitous yet challenging task across various fields, including physics (1), chemistry (2), and economics (3). These distributions are typically of the form $p(\boldsymbol{x}) = \exp\left(-f[\boldsymbol{x}]\right)/Z$, where $f[\cdot]$ is a functional representing, for example, the potential of a chemical compound or the action of a physical system, while $Z$, the normalization constant (or partition function), is often unknown. While $f[\cdot]$ is usually available in closed form, as it describes the microscopic dynamics of the system under study, computing $Z$ would require solving a functional or high-dimensional integral, which is generally intractable. In fact, for many systems of interest, sampling from Boltzmann distributions has been proven to be NP-hard (4), making it highly unlikely that a polynomial-time algorithm exists for this problem. Due to this complexity, sampling from unnormalized Boltzmann distributions is traditionally performed using Markov Chain Monte Carlo (MCMC) methods (5), where a randomly initialized Markov chain is guaranteed to converge to the target distribution. Despite numerous advanced MCMC techniques, significant challenges remain. In chemical and biological systems, for instance, sampling can be hindered by high-energy barriers separating metastable states, posing a major obstacle for tasks such as protein folding (6). In physics, MCMC methods often suffer from slow convergence due to autocorrelations between samples, necessitating longer simulations to obtain statistically independent samples and thereby increasing computational costs (7).

Over the past decade, deep generative models (8) have achieved remarkable success in sampling from Boltzmann distributions within the framework of variational inference (VI) (9). In particular, Ref. (10) introduced Boltzmann Generators (BGs), an approach in which a variational (parametrized) probabil-

---

*Correspondence to jkreit@uni-bonn.de, schuh@hiskp.uni-bonn.de, kim.a.nicoli@gmail.com, and lfuncke@uni-bonn.de

ity density $q_{\boldsymbol{\theta}} \in \mathcal{Q}$ is learned, using a generative model, to approximate the target distribution [1] of a chemical system, i.e., $q_{\boldsymbol{\theta}} \approx p$. Around the same time, concurrent studies proposed similar ideas in the contexts of statistical physics (11; 12) and lattice quantum field theories (13; 14). A distinctive feature of BGs is that they rely on generative models capable of providing the learned variational density in closed form. These include autoregressive neural networks (15; 16) and normalizing flows (NFs) (17; 18), which are particularly suited for sampling discrete and continuous degrees of freedom, respectively. In the remainder of this work, we primarily focus on NFs, although extensions to other generative models that allow exact likelihood computation are also possible.

Despite their potential to overcome some limitations of traditional MCMC sampling, deep generative models present challenges of their own. In particular, to ensure reliable sampling from the target density with suitable asymptotic guarantees (12), these models must first be trained to a sufficiently high standard. Deep generative models, such as NFs, are parametrized by deep neural networks with numerous trainable parameters, which may require a substantial computational effort (training) to converge. To accelerate training, it has been shown that incorporating inductive biases, such as symmetry constraints, into the model architecture can lead to faster and more robust convergence. A seminal example are convolutional neural networks (CNNs), which exhibit built-in translational equivariance (19). This concept has been generalized to arbitrary symmetry groups and manifolds (20; 21; 22). Similar ideas have been extensively leveraged in scientific applications, where chemical and physical systems are often rich in symmetries (23; 24; 25; 26).

In this paper, we propose a general framework for embedding arbitrary symmetries into the training protocol of NFs, which we term Symmetry-Enforcing Stochastic Modulation (SESaMo). Our approach leverages the prior knowledge (symmetries) from the unnormalized log probability to train a NF and uses an independent random variable to infer the correct probability mass for each mode of the target distribution. Crucially, this approach holds promise for mitigating—and potentially overcoming—the fundamental challenge of mode collapse in variational inference (27; 28). We focus on a setting in which no samples from the target distribution are available, and mode collapse is particularly prominent. However, we emphasize that our method is not limited to the data-free case. In summary, the contributions of this work are threefold:

- We propose Symmetry-Enforcing Stochastic Modulation (SESaMo), a novel approach to incorporate continuous and discrete symmetries (broken and exact) into flow-based models.
- We numerically enforce bijectivity by introducing a penalty term in the loss function.
- We conduct extensive numerical experiments to validate our theory on both toy problems and real-world benchmarks for lattice quantum field theories.

The remainder of this paper is organized as follows. In Sec. 2, we introduce the necessary background on NFs and variational inference. We also discuss how symmetries can be incorporated into flow-based models and establish the notation used throughout the manuscript. In Sec. 3, we present our stochastic modulation approach. Finally, in Sec. 4, we validate our approach, both on a standard benchmark and on tasks of practical relevance, such as sampling lattice quantum field theories, including the $\phi^4$ theory and the Hubbard model. In Sec. 5, we discuss the limitations of our algorithm, and we conclude by summarizing our findings and outlining potential directions for future work in Sec. 6.

## 1.1 RELATED WORK

The field of geometric deep learning (29), which investigates the mathematical foundations of deep learning on geometric structures—particularly group-equivariant and gauge-equivariant neural networks—has advanced significantly in recent years. For a comprehensive review of common methodologies, we refer to Refs. (30; 31).

Köhler et al. (32) proposed a way to build NFs that are equivariant under the symmetries of the target $p$, ensuring that the variational distribution $q_{\boldsymbol{\theta}}$ inherently respects these symmetries, thereby improving both accuracy and efficiency. This work laid the foundation for the development of equivariant NFs across various applications. Satorras et al. (33) proposed a generative model equivariant to Euclidean

---

[1]For notational convenience, we use the same symbol for a distribution and its density with respect to the Lebesgue measure.

symmetries, integrating E(n)-Equivariant Graph Neural Networks (EGNNs) (25) within a continuous NF framework (34), yielding an invertible map that preserves Euclidean invariances. Bose et al. (35) addressed the general problem of constructing equivariant diffeomorphisms with an equivariant *finite* NF, specifically targeting finite symmetry groups and compact spaces. In high-energy physics, Kanwar et al. (36) and Boyda et al. (37) adapted NFs to respect Abelian and non-Abelian gauge symmetries, respectively. In condensed matter physics, Schuh et al. (38) demonstrated the importance of enforcing equivariance in NFs for symmetry-rich systems like the Hubbard model, showing that equivariance is crucial for accurately learning the target density and overcoming ergodicity issues. For atomistic systems (39) and atomic solids (40), Wirnsberger et al. introduced NFs equipped with permutation-equivariant diffeomorphisms. More recently, Midgley et al. (41) introduced NFs that inherently respects SE(3) group symmetries—comprising translations, rotations, and reflections—as well as permutation invariance. Furthermore, Klein et al. (42) proposed equivariant flow matching, a training objective based on optimal transport flow matching that leverages inherent symmetries in physical systems, enabling simulation-free training of equivariant continuous normalizing flows (CNFs). In the context of diffusion models (43), Hoogeboom et al. (44) introduced an E(3)-equivariant diffusion model for 3D molecular generation, which, similar to (33), enforces Euclidean invariance under translations and rotations. Lastly, Pires et al. (45) showed that a Variational Mixture of NFs (VMoNF) can model the symmetries of labelled data.

## 2 Preliminaries

### 2.1 Normalizing Flows

Normalizing flows (NFs) (18) are a class of generative models that provide an effective framework for approximating complicated probability distributions. Commonly employed in the context of variational inference (VI) (46), NFs operate by transforming a simple, well-understood, prior distribution (typically a Gaussian) into a target distribution through a sequence of invertible and differentiable mappings. A key advantage of NFs is their ability to efficiently sample from approximated high-dimensional distributions while retaining the capability to compute exact likelihoods. This exact likelihood computation distinguishes NFs from many other generative models, making them particularly well-suited for learning probability distributions in scientific applications, such as chemistry (10) and physics (14). NFs can be categorized based on how the mappings between the prior density and the target distribution are constructed. These categories include coupling-based NFs (47; 48; 49), autoregressive NFs (50), and continuous NFs (34). For the sake of simplicity, this paper primarily focuses on coupling-based NFs, although extensions to other types of NFs are possible.

At the heart of NFs lies the concept of a *bijective transformation* that maps samples from a prior distribution $z \sim q_0(z)$ (such as a multivariate Gaussian) to samples from a variational distribution $x \sim q_\theta$, which is meant to approximate a target $p$. This typically happens by means of a learnable function

$$g_\theta : z \sim q_0 \to x \sim q_\theta(x) \,, \tag{1}$$

where the transformation $g_\theta$ is parametrized by a neural network. To increase the flexibility of NFs, multiple transformations (coupling blocks) can be composed, allowing for more expressive mappings between prior and target distributions, $g_\theta(z) = g_{\theta_T} \circ g_{\theta_{T-1}} \circ \ldots \circ g_{\theta_1}(z)$. A key feature of NFs is that the transformation must be *invertible*, allowing the likelihood of the target distribution to be computed exactly using the change of variables formula

$$q_\theta(x) = q_0(g_\theta^{-1}(x)) \left| \det \left( \frac{\partial g_\theta^{-1}(x)}{\partial x} \right) \right| \,. \tag{2}$$

For a comprehensive overview of NFs, we refer to the review papers (18; 51). In this work, we focus mainly on affine NF architectures, such as RealNVP (48) and NICE (47).

### 2.2 The Kullback-Leibler Divergence

In the context of Variational Inference, the parameters $\theta$ of NFs are trained by minimizing the so-called (reverse) Kullback-Leibler (KL) divergence (52)

$$\mathrm{KL}(q_\theta \,\|\, p) = -\mathbb{E}_{x \sim q_\theta} \left[ \ln \frac{\tilde{p}(x)}{q_\theta(x)} \right] + \ln Z \,, \tag{3}$$

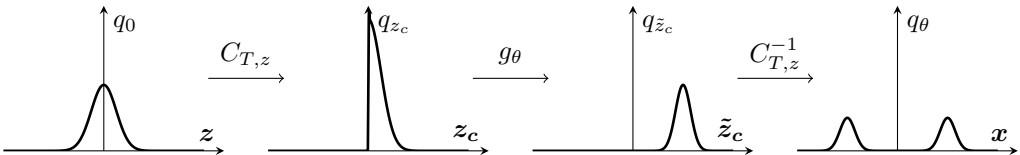

Figure 1: Visualization of the *canonicalization* approach making a flow-based model equivariant with respect to a $\mathbb{Z}_2$ symmetry.

where $\tilde{p}(\boldsymbol{x}) = \exp\left(-f[\boldsymbol{x}]\right)$ and $q_{\boldsymbol{\theta}}(\boldsymbol{x})$ are the unnormalized target and the parametrized probability distributions, respectively. The logarithm of the unknown partition function simply appears as an additive term, which vanishes upon taking the gradient. It should also be noted that the KL divergence is not symmetric, i.e., $\mathrm{KL}(q_{\boldsymbol{\theta}} \,\|\, p) \neq \mathrm{KL}(p \,\|\, q_{\boldsymbol{\theta}})$. Consequently, training using Eq. (3) differs from the practice of maximum likelihood training, which employs the *forward* KL-divergence—a common approach in, e.g., computer vision applications (50). This distinction is significant: In Variational Inference, access to training data is often unavailable, and models must be trained solely using the closed-form unnormalized log probability $\tilde{p}$, which is the focus of this work.

## 2.3 Equivariant Normalizing Flows

In previous works, several attempts have been made to incorporate prior knowledge into NFs and make them equivariant with respect to certain symmetry groups. The main result stemming from (32) is summarised in the following theorem:

**Theorem 1 (Köhler et al., (2020))** *Let's assume $H$ is a group acting on $\mathbb{R}^n$, $q_0$ is the base density of a flow-based transformation with $q_{\boldsymbol{\theta}}$ being the transformed density under the diffeomorphism $g_{\boldsymbol{\theta}} : \mathbb{R}^n \to \mathbb{R}^n$. If $g_{\boldsymbol{\theta}}$ is an $H$-equivariant diffeomorphism and $q_0$ is an $H$-invariant density with respect to the same group $H$, then $q_{\boldsymbol{\theta}}$ is also an $H$-invariant density on $\mathbb{R}^n$.*

Specifically, this theorem provides a general protocol to build an *equivariant* NF by choosing an appropriate invertible map $g_{\boldsymbol{\theta}}$ that is $H$-equivariant. However, despite the generality of this result, defining equivariant diffeomorphisms that allow for tractable inverses and Jacobians—both essential for building an NF—remains an open challenge. Indeed, different approaches have been leveraged in recent works to build equivariant flow-based models.

### 2.3.1 Equivariant Neural Networks

In coupling-based NFs, the diffeomorphism $g_{\boldsymbol{\theta}}$ is often parametrized by a neural network (NN). A straightforward approach to enforce equivariance (or invariance) (53; 54) is to design an NN that explicitly satisfies these symmetry requirements. However, a significant limitation of this method is that constructing such constrained architectures is neither always possible nor straightforward. One instance where this approach is feasible is in the case of a $\mathbb{Z}_2$ symmetry. Indeed, recent work showed how to build manifestly sign-equivariant architectures (55). For example, a simple strategy to achieve sign equivariance in NNs is to use equivariant activation functions, such as *tanh*, and omit bias terms, ensuring that the resulting NN remains equivariant. Indeed, this approach was successfully applied for training $\mathbb{Z}_2$-equivariant NFs in the context of lattice quantum field theories (14; 56; 57).

### 2.3.2 Canonicalization

The idea of *canonicalization*, largely motivated by Theorem 1, has been widely explored in the context of flow-based sampling for lattice field theories (36). Indeed, physical systems are rich in global (and local) symmetries, and being able to develop equivariant flows fulfilling these constraints is a very active area of research. The key idea is to use a transformation $C_{T,z}$ to map a sample from the base density to a so-called canonical cell $\Omega$, see (37). The NF then transforms the canonicalized sample, before the inverse $C_{T,z}^{-1}$ is applied to map the sample back to its original space. We refer to Fig. 3 and App. B for more details. A parametric map $g_{\boldsymbol{\theta}}$ is equivariant to a generic transformation $T$ if

$$g_{\boldsymbol{\theta}}(T\boldsymbol{x}) = Tg_{\boldsymbol{\theta}}(\boldsymbol{x}) \implies g_{\boldsymbol{\theta}}(\boldsymbol{x}) = T^{-1}g_{\boldsymbol{\theta}}(T\boldsymbol{x}). \tag{4}$$

For example, for the sign-flipping $\mathbb{Z}_2$ transformation mentioned above, the transformation reads

$$T_{\mathbb{Z}_2} : \boldsymbol{x} \to -\boldsymbol{x} \,. \tag{5}$$

A canonical map $C_{T,z}$ transforms samples $\boldsymbol{z} \in \mathbb{R}^n$, where $\boldsymbol{z} \sim q_0$, to the canonical $\Omega$

$$C_{T,z} : \boldsymbol{z} \in \mathbb{R}^n \to \boldsymbol{z}_c \in \Omega \qquad \text{with the inverse} \qquad C_{T,z}^{-1} : \widetilde{\boldsymbol{z}}_c \in \widetilde{\Omega} \to \boldsymbol{x} \in \mathbb{R}^n. \tag{6}$$

The two manifolds $\Omega$ and $\widetilde{\Omega}$ are connected by the diffeomorphism $g_{\boldsymbol{\theta}}$ acting in the *canonical space*, i.e.,

$$g_{\boldsymbol{\theta}} : \boldsymbol{z}_c \in \Omega \to \widetilde{\boldsymbol{z}}_c \in \widetilde{\Omega} \qquad \text{and} \qquad g_{\boldsymbol{\theta}}^{-1} : \widetilde{\boldsymbol{z}}_c \in \widetilde{\Omega} \to \boldsymbol{z}_c \in \Omega \,. \tag{7}$$

Note that $C_{T,z}$ depends on some *specific* symmetry transformation $T$, see Eq. (4), which makes the *canonicalized flow* $\widetilde{g}_{\boldsymbol{\theta}}(\boldsymbol{z}) = C_{T,z}^{-1} g_{\boldsymbol{\theta}} C_{T,z}(\boldsymbol{z})$ equivariant. Focusing on the sign-flipping $\mathbb{Z}_2$ transformation mentioned above, we have

$$C_{T,z} : \boldsymbol{z} \mapsto \begin{cases} \boldsymbol{z}, & \text{if } \boldsymbol{z} \in \Omega \\ -\boldsymbol{z}, & \text{else} \end{cases} \tag{8}$$

where the canonical cell in this case is $\Omega = \{\boldsymbol{z} \in \mathbb{R}^n \text{ s.t. } \sum_{i=1}^n z_i \geq 0\}$. See Fig. 1 for a visual intuition. This approach can be generalised for $\boldsymbol{z} \in \mathbb{R}^n$ and a set of $M$ symmetry transformations $\{T_i\}$ such that

$$C_{\mathbf{T},z} : \boldsymbol{z} \mapsto \begin{cases} \boldsymbol{z}, & \text{if } \boldsymbol{z} \in \Omega \\ T_1 \boldsymbol{z}, & \text{elif } \boldsymbol{z} \in A_1 \\ \vdots \\ T_M \boldsymbol{z}, & \text{elif } \boldsymbol{z} \in A_M \end{cases} \quad \text{with inverse} \quad C_{\mathbf{T},z}^{-1} : \boldsymbol{x} \mapsto \begin{cases} \boldsymbol{x}, & \text{if } \boldsymbol{z} \in \Omega \\ T_1^{-1} \boldsymbol{x}, & \text{elif } \boldsymbol{z} \in A_1 \\ \vdots \\ T_M^{-1} \boldsymbol{x}, & \text{elif } \boldsymbol{z} \in A_M \end{cases} \tag{9}$$

where $A_i$ denote different regions of the target space, for example corresponding to different modes of the target distribution. Note that the inverse transformation $C_{\mathbf{T},z}^{-1}$ depends on the input $\boldsymbol{z}$, i.e., the information about the origin of the sample in the base space must be retained in the transformation. A proof that the canonicalization approach is equivariant is given in App. C.

### 2.3.3 CONSTRAINTS ON CANONICALIZATION

In order to enforce equivariance via canonicalization, two constraints must be met: first the prior distribution $q_0$ must be *invariant* under any symmetry transformation $T_i$ (32), i.e., $q_0(\boldsymbol{z}) = q_0(T_i \boldsymbol{z})$. Second, $g_{\boldsymbol{\theta}}$ should not map samples *outside* of the canonical cell, i.e., $\tilde{\boldsymbol{z}}_c = g_{\boldsymbol{\theta}}(C_{\mathbf{T},z} \boldsymbol{z}) \in \Omega$. While the former constraint can be readily verified, the latter may not hold for any general NF. We enforce this latter constraint by introducing a regularization term

$$\Lambda(\tilde{\boldsymbol{z}}_c) = A \cdot \sigma(B \cdot \lambda(\tilde{\boldsymbol{z}}_c)) \cdot \Theta(\lambda(\tilde{\boldsymbol{z}}_c)) \,, \tag{10}$$

where $\lambda(\tilde{\boldsymbol{z}}_c)$ is a *penalty function* being zero for a general input $\tilde{\boldsymbol{z}}_c$ at the boundary $\partial\Omega$ of the canonical cell $\Omega$, negative for $\tilde{\boldsymbol{z}}_c \in \Omega$, and positive for $\tilde{\boldsymbol{z}}_c \notin \Omega$. The Heaviside step function $\Theta(\cdot)$ ensures that the penalty term is zero for $\tilde{\boldsymbol{z}}_c \in \Omega$, while the sigmoid function $\sigma(\cdot)$ ensures that the penalty function has a gradient pointing toward the canonical cell $\Omega$. The hyperparameters $A, B \in \mathbb{R}$ are used to scale the amplitude and the gradient of the function, respectively. While $A$ needs to match at least the order of magnitude of the loss function, $B$ must be chosen sufficiently small to ensure non-vanishing gradients for $\tilde{\boldsymbol{z}}_c \notin \Omega$. This regularization term is added to Eq. (3) during the training of a NF. We provide further details about the penalty term in App. D.

## 3 PROPOSED METHOD: SESAMO

Crucially, certain symmetries may be difficult to incorporate through naive canonicalization strategies and are unlikely to be effectively captured by standard flow-based generative models. A representative case is a one-dimensional multimodal distribution with modes of unequal probability mass (see App. E and App. H). Our proposed method, Symmetry-Enforcing Stochastic Modulation (SESaMo), introduces a novel stochastic modulation mechanism that is described in detail in Sec. 3.1.

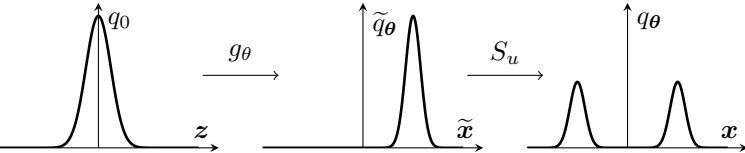

Figure 2: Visualization of the *stochastic modulation* approach for enforcing a $\mathbb{Z}_2$ symmetry in a flow-based model.

## 3.1 STOCHASTIC MODULATION

Stochastic modulation involves drawing samples $\widetilde{x}$ from a flow-based sampler with density $\widetilde{q}_{\theta}(\widetilde{x})$. Prior samples $z \sim q_0$ (left panel in Fig. 2) are shifted by the flow $g_{\theta}$ into the canonical cell $\Omega$ to align with one of the modes, producing $\widetilde{x} \sim \widetilde{q}_{\theta}$ with $\widetilde{x} \in \Omega$ (center panel). This shift is induced by the penalty term $\Lambda(\widetilde{x})$ in Eq. (10). The samples $\widetilde{x}$ are then transformed according to the stochastic modulation $S_u$ (right panel), which is conditioned on a random variable $u$, resulting in an output density

$$q_{\theta,b}(x) = \sum_u p_{S,b}(u) \cdot q_0\left(\widetilde{g}_{\theta,u}^{-1}(x)\right) \cdot \left| \det\left(\frac{\partial \widetilde{g}_{\theta,u}^{-1}}{\partial x}\right) \right| . \tag{11}$$

Here, $p_{S,b}(u)$ is the modulation probability that depends on learnable parameters $b$, whose number is fixed by the symmetry type. The diffeomorphic map from the base density $q_0(z)$ to the final density reads

$$\widetilde{g}_{\theta,u}(z) = S_u(g_{\theta}(z)), \tag{12}$$

such that

$$\det\left(\frac{\partial \widetilde{g}_{\theta,u}}{\partial z}\right) = \det\left(\frac{\partial S_u}{\partial g_{\theta}}\right) \det\left(\frac{\partial g_{\theta}}{\partial z}\right). \tag{13}$$

A general stochastic modulation $S_{\mathbf{T},u}$ for a set of $M$ transformations $\{T_i\}$ reads

$$S_{\mathbf{T},u} : x \mapsto \begin{cases} x, & \text{if } u = 0 \\ T_1 x, & \text{elif } u = 1 \\ \vdots & \\ T_M x, & \text{elif } u = M \end{cases} \quad \text{with} \quad u \sim p_{S,b} \tag{14}$$

where the transformations $T_i$ map samples $x \sim q_{\theta}(x)$ to distinct regions in the configuration space, potentially corresponding to different modes of the target distribution $p(x)$. We note that $T_i \neq T_j, \forall i \neq j$ and that $T_i x \notin \Omega$, in order to avoid any overlap between distributions and thus preserve bijectivity.

From Eqs. (11)–(13), if $u \sim p_{S,b}(u)$ is sampled and not marginalized, we can write the log probability as

$$\ln q_{\theta}(\widetilde{g}_{\theta,u}(z)) = \ln p_{S,b}(u) + \ln q_0(z) - \ln\left|\det \frac{\partial S_u}{\partial g_{\theta}}\right| - \ln\left|\det \frac{\partial g_{\theta}}{\partial z}\right|, \tag{15}$$

where $z = \widetilde{g}_{\theta,u}^{-1}(x)$ and $p_{S,b}(u)$ is the probability of sampling the random variable $u$. To better understand this mechanism, let us again consider a target density with $\mathbb{Z}_2$ symmetry. In this specific case, we define a random variable $u \in \{0, 1\}$ that follows a Bernoulli distribution $\mathcal{B}(e^b)$ and

$$S_u : x \rightarrow \begin{cases} x & \text{if } u = 0 \\ -x & \text{if } u = 1 \end{cases} \quad \text{with} \quad u \sim p_{S,b} = \mathcal{B}(e^b) \quad \text{and} \quad b = \ln 0.5 . \tag{16}$$

For a broken $\mathbb{Z}_2$ symmetry on the other hand, the modulation probability is given in terms of a learnable parameter $b \in \mathbb{R}^-$ and reads

$$p_{S,b} = \begin{cases} 1 - e^b & \text{if } u = 0 \\ e^b & \text{if } u = 1 . \end{cases} \tag{17}$$

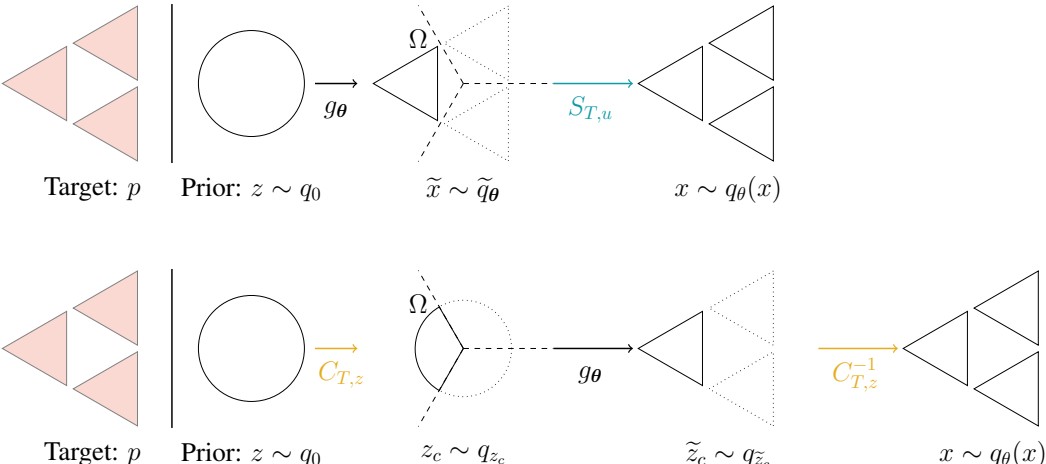

Figure 3: Illustration of Symmetry-Enforcing Stochastic Modulation (SESaMo) (top row) and canonicalization (bottom row), shown for an example target distribution and corresponding prior.

Unlike canonicalization, SESaMo (illustrated in the upper row of Fig. 3) employs a flow $g_{\boldsymbol{\theta}}$ that shifts the prior density $q_0$ so that it captures a single mode of the target density within the canonical cell. This shift is induced by the penalty term $\Lambda(\widetilde{\boldsymbol{x}})$. After this alignment step, the probability mass is redistributed according to $S_u$. In contrast, canonicalization (shown in the lower row of Fig. 3) applies a fixed transformation $C_{T,z}$ that maps $z$ directly into the canonical cell. Consequently, unlike canonicalization, SESaMo does not require the prior $q_0$ to be invariant under the symmetry transformation, which makes it applicable in a broader range of settings.

Similarly to canonicalization, stochastic modulation requires the map $S_u$ to be bijective. This property is enforced by the penalty term introduced in Sec. 2.3.3. An extended comparison between canonicalization and stochastic modulation is provided in App.B. A pseudocode representation of SESaMo can be found in Sec. A.

If the probability mass is not evenly distributed among the modes of the target density (e.g., $b \neq \ln 0.5$ for a $\mathbb{Z}_2$ symmetry), having a learnable parameter $b$ allows SESaMo to effectively capture the broken symmetry. This case is further detailed in App. E. We also stress that SESaMo extends beyond discrete symmetries. Appendix F develops a formulation for (broken) continuous symmetries, where $u$ is a continuous variable instead of a discrete one.

### 3.2 LOSS FUNCTION AND REINFORCE ESTIMATOR

For training, the standard reverse KL divergence is extended by the penalty term $\Lambda(\cdot)$

$$\widetilde{\mathrm{KL}}(q_{\boldsymbol{\theta}} \,||\, p) = \mathbb{E}_{\boldsymbol{z} \sim q_0} \mathbb{E}_{u \sim p_{S,b}} \left[ \ln q_{\boldsymbol{\theta}}(\widetilde{g}_{\boldsymbol{\theta},u}(\boldsymbol{z})) + f[\widetilde{g}_{\boldsymbol{\theta},u}(\boldsymbol{z})] + \Lambda(g_{\boldsymbol{\theta}}(\boldsymbol{z})) \right] , \tag{18}$$

stemming from Sec. 2.3.3. Since the parameter $b$ of the stochastic modulation is only present in the modulation probability $p_{S,b}$ the REINFORCE estimator (58) is used to enable gradient computation through the random variable $u$

$$\frac{\partial}{\partial b} \mathbb{E}_{u \sim p_{S,b}} \left[ \ln q_{\boldsymbol{\theta}}(\widetilde{g}_{\boldsymbol{\theta},u}(\boldsymbol{z})) + f[\widetilde{g}_{\boldsymbol{\theta},u}(\boldsymbol{z})] \right] = \mathbb{E}_{u \sim p_{S,b}} \left[ (\ln q_{\boldsymbol{\theta}}(\widetilde{g}_{\boldsymbol{\theta},u}(\boldsymbol{z})) + f[\widetilde{g}_{\boldsymbol{\theta},u}(\boldsymbol{z})]) \cdot \frac{\partial}{\partial b} \ln p_{S,b}(u) \right] . \tag{19}$$

## 4 NUMERICAL EXPERIMENTS

In this section, we present numerical experiments that compare the performance of four approaches: Flow Annealed Importance Sampling Bootstrap (FAB) (59), RealNVP with Variational Mixture of Normalizing Flows (VMoNF) (45), RealNVP with canonicalization (37), and RealNVP with stochastic modulation (SESaMo). We benchmark these approaches both on toy problems and on

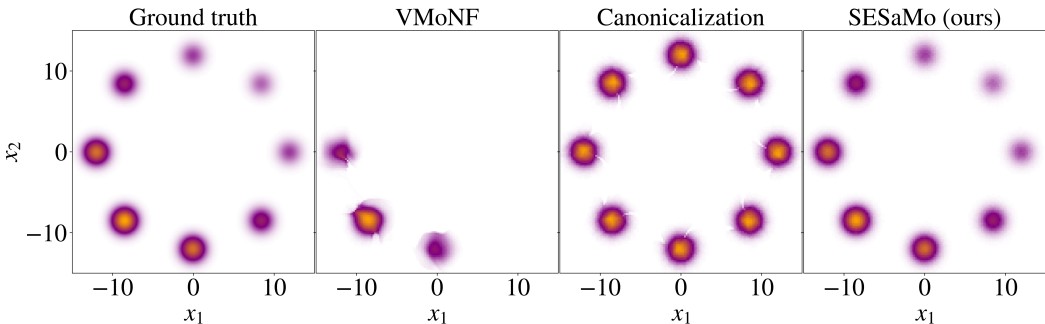

Figure 4: Gaussian mixture target density (broken $\mathbb{Z}_8$ symmetry). All flow-based models are trained until convergence. From left to right we show: the ground truth, VMoNF, canonicalization, and SESaMo (ours). We refer to App. I for more details on the experiments.

physically relevant tasks. To evaluate the effectiveness of each approach, we use the the KL divergence (if the normalization $Z$ is known) and the effective sample size (ESS) (12), as a performance metric. The ESS quantifies the accuracy with which the approximation $q_\theta$ matches the target probability distribution $p$. Bounded between zero and one, the ESS reaches its optimal value (ESS = 1) when the approximation is exact ($q_\theta = p$). Although the ESS is not a good metric if the distribution lacks full mode coverage, the symmetry-enforcing architecture guarantees us to capture all modes, making the ESS a valid metric to quantify SESaMo's performance.

The code used to run these experiments builds on an earlier release of (60). A live version is available at github.com/fifi-research/sesamo, and a static, archived snapshot is provided in Ref. (61).

### 4.1 Toy Example: Gaussian Mixture

We initially consider a probability distribution in two dimensions whose density is given by

$$p(\boldsymbol{x}) = \frac{1}{Z} \sum_{k=1}^{N} \exp\left(\frac{-(\boldsymbol{x} - \boldsymbol{\mu}_k)^2}{2} - \alpha(x_1 + x_2)\right), \quad \boldsymbol{\mu}_k = R \cdot \left(\cos\left(\frac{2\pi k}{N}\right), \sin\left(\frac{2\pi k}{N}\right)\right)^T,$$

(20)

where $Z = 2\pi \sum_{k=1}^{N} \exp\left(\alpha^2 - \alpha R\sqrt{2}\sin(\frac{2\pi k}{N} + \frac{\pi}{4})\right)$ is the normalization, $N \in \mathbb{N}^+$ is the number of Gaussians, $R \in \mathbb{R}^+$ is the radius of the circle around which they are located, and $\alpha \in \mathbb{R}$ breaks the $\mathbb{Z}_N$ symmetry of this model. In this study, we use $N = 8$, $R = 12$, and $\alpha = 0$ ($\alpha = 0.05$), which results in a (broken) $\mathbb{Z}_8$ symmetry. In Tab. 1, we report the ESS and KL divergence achieved after convergence, and we visualize the corresponding target density in Fig. 4. Even though VMoNF has the ability that different NFs learn different sectors of the target distribution, it collapses to the three most likely modes in the lower left. This is not an issue of the architecture itself but the mode collapsing behaviour of the reverse KL that is used in this work. Overall, SESaMo achieves the best performance, outperforming the other baselines and yielding higher accuracy. For more details we refer to App. H.

### 4.2 Physics Example: Lattice Quantum Field Theory

Sampling using NFs has become ubiquitous across various fields of physics, yielding particularly notable results for sampling lattice quantum chromodynamics (62), scalar lattice quantum field theories (14; 63), and condensed matter systems (38). We refer to (64) for a comprehensive overview. In what follows, we primarily focus on two pertinent benchmarks: the complex $\phi^4$ theory and the Hubbard model. We direct readers seeking further technical details regarding the physics to App. G.

In lattice quantum field theory, the probability distribution of a system is given by a Boltzmann-like density $p(\boldsymbol{x}) = \exp(-f[\boldsymbol{x}])/Z$, where $f[\boldsymbol{x}]$ is a functional known as the *action*, $Z$ is an unknown partition function, and $\boldsymbol{x}$ denotes the lattice fields. Note that, as discussed in Sec. 3, for the following experiments we optimize the symmetry breaking parameter $b$ during training, which, as shown in App. H, perfectly agrees with the analytical prediction.

**The complex $\phi^4$ scalar field theory in two dimensions**    The complex $\phi^4$ theory offers a simple yet versatile framework for investigating interacting scalar fields. It plays a crucial role in understanding spontaneous symmetry breaking (including the Higgs mechanism) and critical phenomena (65), while providing a key testbed for the machine learning community to develop theoretical techniques and numerical methods (66; 67; 68). We consider the action with quartic interactions,

$$f[\boldsymbol{x}] = \sum_{j \in V} \left[ -2\kappa \sum_{\hat{\mu}=1}^{2} (\boldsymbol{x}_j \boldsymbol{x}_{j+\hat{\mu}}) + (1-2\lambda)\boldsymbol{x}_j^2 + \lambda \boldsymbol{x}_j^4 + \alpha \mathrm{Re}[\boldsymbol{x}_j] \right], \qquad (21)$$

where $\boldsymbol{x} = \boldsymbol{x}_1 + i\boldsymbol{x}_2$ are the complex scalar fields, the subscript $j$ labels the lattice sites in the two-dimensional lattice volume $V = N_x \times N_t = 8 \times 8$, the $\kappa$ and $\lambda$ are the couplings of the theory, and $\hat{\mu}$ denotes the interactions between nearest neighbours. The term $\alpha \mathrm{Re}(\boldsymbol{x})$ introduces an additional component designed to break the continuous $U(1)$ symmetry of the theory, thereby increasing the complexity of the learning task.[2] We emphasize that while prior studies have often focused on *real* scalar fields, physical fields are complex-valued. Therefore, we compare SESaMo with FAB and VMoNF when sampling $\boldsymbol{x} \in \mathbb{C}^n$. Canonicalization could not be applied here. The ESS obtained by each model is detailed in Tab. 1 for both broken ($\alpha \neq 0$) and unbroken ($\alpha = 0$) $U(1)$ symmetry. Across both conditions, SESaMo achieved the highest ESS, indicating its superior ability to incorporate the underlying physical symmetries into the flow model. Additional results, including the density plots, are available in App. H. Moreover, App. H also demonstrates how SESaMo outperforms the baselines of RealNVP and canonicalization in the case of *real* scalar field theory.

**The Hubbard model in two dimensions**    The Hubbard model is a cornerstone of condensed matter physics, providing a fundamental description of interacting electrons on a lattice and playing a pivotal role in studying phenomena such as magnetism, metal-insulator transitions, and high-temperature superconductivity (70). For our numerical experiments, we adopt the setup as detailed in (38; 71), with the action—featuring a broken $\mathbb{Z}_4$ symmetry—given by

$$f[\boldsymbol{x}] = \frac{1}{2\widetilde{U}} \sum_{j \in V} \boldsymbol{x}_j^2 - \log \det M[\boldsymbol{x}] - \log \det M[-\boldsymbol{x}], \qquad (22)$$

where the coupling $\widetilde{U}$ describes the interaction strength, $M[\cdot]$ is the *fermion matrix* describing the interacting fermions (particles), $\boldsymbol{x}$ are auxiliary bosonic fields, and the subscript $j$ labels the lattice sites in the lattice volume $V = N_x \times N_t$. While for the GMM and the complex $\phi^4$ theory we focused on small lattice volumes as a proof-of-principle demonstration, here we include both a small volume of $V = 2 \times 1$ to compare with analytical solutions and a large volume of $V = 18 \times 100$ to demonstrate the scalability of our approach. We refer to Apps. E, G, H, and (38) for more details about the model. For learning the Boltzmann distribution, we again compare FAB, VMoNF, canonicalization, and SESaMo. For the smaller volume of $V = 2 \times 1$, the ESS and KL divergence are reported in Tab. 1, and the resulting probability density after training is shown in Fig. 5. For the larger volume of $V = 18 \times 100$, the ESS is also listed in Tab. 1, whereas the KL divergence cannot be reported due to the unknown normalization of the target distribution. FAB is not feasible at this lattice volume. As before, SESaMo achieves the highest ESS and exhibits faster and more stable convergence compared to the other baselines. For further results and density plots illustrating that SESaMo mitigates mode collapse (72), we refer the reader to App. H. While Schuh et al. (38) first demonstrated the application of NFs to the Hubbard model using canonicalization and small lattice volumes of $V = 2 \times 2$, SESaMo with the objective in Eq. (18) crucially achieves a higher ESS, perfectly learns the broken $\mathbb{Z}_4$ symmetry, and scales to lattice volumes of up to $V = 18 \times 100$ (broken $(\mathbb{Z}_2)^{18}$ symmetry), thereby establishing a new state-of-the-art.

## 5    LIMITATIONS

A primary limitation of SESaMo stems from the requirement that the symmetry sectors must be known a priori to apply the stochastic modulation. Nevertheless, for applications in physics and

---

[2]This system can serve as a proxy to describe a quantum field theory with two flavors of differing masses (69).

[3]The reverse KL divergence can only be computed if the normalization of the probability distribution is known, which is the case for the GMM and the Hubbard model with $V = 2 \times 1$.

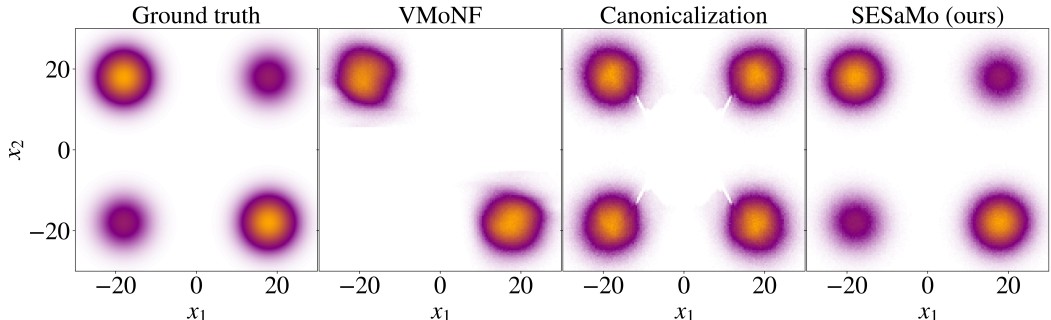

Figure 5: Density for the Hubbard model (broken $\mathbb{Z}_4$ symmetry) for $V = N_x \times N_t = 2 \times 1$. All flow-based models are trained until convergence. From left to right we show: the ground truth, VMoNF, canonicalization, and SESaMo (ours). We refer to App. I for more details on the experiments.

| | Model | Volume | Symmetry | FAB | VMoNF | Canon. | SESaMo |
|---|---|---|---|---|---|---|---|
| **ESS** | GMM | $2 \times 1$ | exact $\mathbb{Z}_8$ | 0.78(3) | 0.61(1) | 0.91(8) | **0.9986(2)** |
| | GMM | $2 \times 1$ | broken $\mathbb{Z}_8$ | 0.81(1) | 0.83(11) | 0.747(2) | **0.9947(3)** |
| | $\phi^4$ theory | $8 \times 8$ | exact $U(1)$ | 0.26(3) | 0.22(2) | – | **0.9472(8)** |
| | $\phi^4$ theory | $8 \times 8$ | broken $U(1)$ | 0.28(5) | 0.23(1) | – | **0.941(2)** |
| | Hubbard | $2 \times 1$ | broken $\mathbb{Z}_4$ | 0.946(9) | 0.37(12) | 0.839(5) | **0.996(1)** |
| | Hubbard | $18 \times 100$ | broken $(\mathbb{Z}_2)^{18}$ | 0.06(5) | – | 0.024(1) | **0.74(1)** |
| **KL** | GMM | $2 \times 1$ | exact $\mathbb{Z}_8$ | 1.19(37) | 0.79(11) | 0.013(2) | **0.0008(1)** |
| | GMM | $2 \times 1$ | broken $\mathbb{Z}_8$ | 0.84(26) | 1.02(14) | 0.189(3) | **0.0024(2)** |
| | Hubbard | $2 \times 1$ | broken $\mathbb{Z}_4$ | 0.28(8) | 0.74(9) | 0.112(7) | **0.0013(8)** |

Table 1: Effective sample size (ESS, higher is better) and KL divergence[3] (smaller is better) after convergence for different benchmarks. Best results (averages over ten different models) are highlighted in bold.

chemistry, this may not pose a significant problem. Indeed, the well-defined symmetries inherent in many physical and chemical systems often allow for the prior determination of the symmetry sectors, thereby enabling the application of stochastic modulation. Another limitation arises from the penalty term in Eq. (10), which enforces bijectivity of the NF. If the target density assigns non-zero probability at the border of the canonical cell, bijectivity can only be maintained approximately. As a result, the ESS may decrease if only samples that strictly preserve bijectivity are accepted. For example, in the Gaussian mixture model discussed in Sec. 4.1, decreasing the radius $R$ causes the modes to move closer together, thereby increasing the density near the border of the canonical cell. However, in many high-dimensional physics applications, the distance between modes typically increases with the dimensionality of the system, thereby mitigating the impact of bijectivity violations. Nonetheless, we emphasize that these limitations are not specific to SESaMo, but is shared with canonicalization.

## 6 CONCLUSIONS

This paper introduces Symmetry-Enforcing Stochastic Modulation (SESaMo)—a novel and flexible approach for constructing symmetry-enhanced NFs. Moreover, we propose an additional penalty term to the reverse KL divergence to enforce numerical bijectivity. Our extensive numerical experiments demonstrate that stochastic modulation outperforms naïve NFs, mixture models and canonicalization methods. We envision SESaMo as a powerful tool for incorporating inductive biases into generative models when learning target probability densities with challenging symmetries—an essential feature in fields like physics and chemistry. Future work will explore the broader capabilities of SESaMo and assess its potential to achieve state-of-the-art performance not only against generative neural samplers but also relative to established numerical techniques, such as Hamiltonian Monte Carlo.

## REPRODUCIBILITY STATEMENT

We provide our source code under the MIT license at github.com/fifi-research/sesamo. A static, archived version of the code is available in Ref. (61). The repository contains training scripts and instructions to reproduce all main experiments. We specify all hyperparameters in Appendix I and provide default configuration files. Experiments were conducted with Python 3.9 and PyTorch 2.7 on NVIDIA A100 GPUs, but the code runs on other CUDA-enabled GPUs as well.

## ACKNOWLEDGMENTS

The authors thank Luca Johannes Wagner for inspiring discussions during the development of this method and Simran Singh for suggesting to extend SESaMo to continuous symmetries. The authors also thank Shinichi Nakajima and Jan Gerken for useful discussions on earlier versions of this work. The authors gratefully acknowledge the access to the Marvin cluster of the University of Bonn. This project was supported by the Deutsche Forschungsgemeinschaft (DFG, German Research Foundation) as part of the CRC 1639 NuMeriQS – project no. 511713970.

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

## A  PSEUDOCODE

To offer an intuitive overview of our proposed method, we include a pseudocode example in Alg. 1.

---

**Algorithm 1** Training loop of SESaMo

---
1:  Initialize flow $g_{\boldsymbol{\theta}}$ with parameters $\boldsymbol{\theta}$
2:  Initialize stochastic modulation $S_u$ with parameters $b$
3:  **for** iteration = 1 to $N$ **do**
4:      Sample $\boldsymbol{z}^{(1:B)}$ from $q_0$ and evaluate $\ln q_0\left(\boldsymbol{z}^{(1:B)}\right)$
5:      Obtain $\tilde{\boldsymbol{x}}^{(1:B)}$ and $\ln\left|\det\frac{\partial g_{\boldsymbol{\theta}}}{\partial \boldsymbol{z}}\right|$ from flow $\left(\boldsymbol{z}^{(1:B)}\right)$
6:      Compute regularization $\Lambda(\tilde{\boldsymbol{x}})$
7:      Sample $u^{(1:B)}$ from $p_{S,b}$ and evaluate $\ln p_{S,b}(u^{(1:B)})$
8:      Obtain $\boldsymbol{x}^{(1:B)}$ and $\ln\left|\det\frac{\partial S_u}{\partial g_{\boldsymbol{\theta}}}\right|$ from stochastic modulation $S_{u^{(1:B)}}\left(\boldsymbol{x}^{(1:B)}\right)$
9:      Compute unnormalized target probability $f\left(\boldsymbol{x}^{(1:B)}\right)$
10:     Compute log probability $\ln q_{\boldsymbol{\theta}}\left(\boldsymbol{x}^{(1:B)}\right)$
11:     Compute loss $\widetilde{\mathrm{KL}}(q_{\boldsymbol{\theta}}\,\|\,p)\left(\boldsymbol{x}^{(1:B)}\right)$
12:     Update parameters $\theta, b$ according to the REINFORCE estimator
13: **end for**

---

## B  INTUITIVE COMPARISON OF CANONICALIZATION AND STOCHASTIC MODULATION

In the main text, two approaches for effectively incorporating symmetries into generative models such as NFs were introduced: canonicalization in Sec. 2.3.2 and Symmetry-Enforcing Stochastic Modulation (SESaMo) in Sec. 3.1. In this section, we summarize the differences between these approaches on a more intuitive level. To help the reader familiarize with the underlying ideas, we provide an illustration for both SESaMo (top row) and canonicalization (bottom row) in Fig. 3, showing an example target distribution and corresponding prior. In Fig. 3, the goal is to sample from a toy target density $p$ that exhibits three modes, visually represented by the three red triangles on the left of Fig. 3. Both approaches start from a Gaussian prior density $q_0$, represented by a circle.

In the case of SESaMo (top row), a random sample $z \sim q_0$ is transformed by an NF, i.e., a parametric map $g_{\boldsymbol{\theta}}$, such that the probability mass of the prior density is shifted and transformed to cover one of the modes of the target density (depicted as the triangle with a solid black line), which lies within the canonical cell $\Omega$ (dashed black line), while the other symmetric modes (triangles with a dotted black line) remain uncovered. The transformed density is denoted as $\widetilde{q}_{\boldsymbol{\theta}}$. At this stage, the model has captured only one mode of the target density. Subsequently, SESaMo employs the *stochastic modulation* $S_{T,u}$ to redistribute the probability mass towards the other modes of the target density, resulting in the final variational probability distribution $q_{\boldsymbol{\theta}}$. This distribution (visualized by the three triangles) approximates the target density $p$.

The canonicalization approach, depicted in the bottom row of Fig. 3, also starts with a prior Gaussian distribution $q_0$. Samples drawn from the prior distribution are transformed such that any sample $z \sim q_0$ is mapped to the canonical cell $\Omega$ (dashed black line), resulting in the density $q_{z_c}$ (solid black line), while the other symmetric modes (dotted black line) remain uncovered. Subsequently, an NF $g_{\boldsymbol{\theta}}$ learns a bijective map to transform these samples in the canonical space. Canonicalized samples, denoted as $\widetilde{\boldsymbol{x}}_c$, are then drawn from the resulting distribution $\widetilde{q}_{\boldsymbol{\theta}}$, which is illustrated in Fig. 3 as a triangle with a solid black line. Given that the resulting parametrized distribution $\widetilde{q}_{\boldsymbol{\theta}}$ is in the canonical space, it needs to be transformed back to the input space. This is achieved by applying the inverse of the initial transformation $C_{T,z}^{-1}$ to the samples $\widetilde{x}_c$, resulting in the final parametrized probability distribution $q_{\boldsymbol{\theta}}$. The support of $q_{\boldsymbol{\theta}}$ is visualized in the right-most plot of the bottom row by three triangles that approximate the target density $p$.

## C  EQUIVARIANCE OF THE CANONICALIZATION METHOD

Let us consider a general symmetry transformation $T$ under which some function $\xi(\cdot) : \boldsymbol{x} \in \mathbb{R}^n \to \xi(\boldsymbol{x}) \in \mathbb{R}$ is invariant, i.e., $\xi(\boldsymbol{x}) = \xi(T\boldsymbol{x})$. A concrete example of such a function can be the action of a physical system, such as Eqs. (21) and (22). A learnable map $g_{\boldsymbol{\theta}} : \boldsymbol{z} \in \Omega \to \widetilde{\boldsymbol{z}} \in \widetilde{\Omega}$ is *equivariant under $T$*, and is thus denoted $\widetilde{g}_{\boldsymbol{\theta}}$, if it satisfies the following condition:

$$\widetilde{g}_{\boldsymbol{\theta}}(T\boldsymbol{z}) = T\widetilde{g}_{\boldsymbol{\theta}}(\boldsymbol{z}). \tag{23}$$

The canonicalization approach, introduced in Sec. 2.3.2, leverages a so-called *canonical transformation* $C_{T,z} : \mathbb{R}^n \to \Omega$ to map samples from the input space into the canonical cell $\Omega$, thereby making the map $\widetilde{g}_{\boldsymbol{\theta}}$ equivariant with respect to $T$. The equivariant map $\widetilde{g}_{\boldsymbol{\theta}}$ thus reads

$$\widetilde{g}_{\boldsymbol{\theta}}(\boldsymbol{z}) = C_{T,z}^{-1} \, g_{\boldsymbol{\theta}}(C_{T,z}\boldsymbol{z}), \tag{24}$$

where $C_{T,z}$ maps a sample $\boldsymbol{z}$ into the canonical cell $\Omega$, $g_{\boldsymbol{\theta}}$ denotes a specific NF, and $C_{T,z}^{-1}$ maps the canonicalized (and transformed) sample $\widetilde{\boldsymbol{z}} = g_{\boldsymbol{\theta}}(C_{T,z}\boldsymbol{z})$ back to the original input space.

In this section, we restrict ourselves to involutory symmetry transformations, i.e., $T^2 = \mathbb{1}$. Our goal is thus to show that canonicalization fulfils the equivariant condition in Eq. (23). We define the canonical transformation

$$C_{T,z} : \boldsymbol{z} \mapsto \begin{cases} \boldsymbol{z}, & \text{if } \boldsymbol{z} \in \Omega \\ T\boldsymbol{z}, & \text{if } T\boldsymbol{z} \in \Omega, \end{cases} \tag{25}$$

with the inverse transformation

$$C_{T,z}^{-1} : \boldsymbol{x} \mapsto \begin{cases} \boldsymbol{x}, & \text{if } \boldsymbol{z} \in \Omega \\ T\boldsymbol{x}, & \text{if } T\boldsymbol{z} \in \Omega. \end{cases} \tag{26}$$

It is crucial to note that the inverse transformation $C_{T,z}^{-1}$ still depends on the sample $\boldsymbol{z}$ to which the canonical transformation $C_{T,z}$ was initially applied, i.e., the information about the initial sample $\boldsymbol{z}$ is implicitly stored in the transformation. One way to check if the map Eq. (24) is *really* equivariant under the transformation $T$ is to sequentially apply the transformation $T$ and then $C_{T,z}$ to the input $\boldsymbol{z}$,

$$C_{T,Tz} : T\boldsymbol{z} \mapsto \begin{cases} T\boldsymbol{z}, & \text{if } T\boldsymbol{z} \in \Omega \\ TT\boldsymbol{z}, & \text{if } TT\boldsymbol{z} \in \Omega \end{cases} = \begin{cases} T\boldsymbol{z}, & \text{if } T\boldsymbol{z} \in \Omega \\ \boldsymbol{z}, & \text{if } \boldsymbol{z} \in \Omega \end{cases}. \tag{27}$$

Note that the involutory property $TT = \mathbb{1}$ has been used here.[4] It follows that the transformations $C_{T,Tz}$ and $C_{T,z}$ are equivalent,

$$C_{T,Tz} \, T\boldsymbol{z} = C_{T,z} \, \boldsymbol{z} \,, \tag{28}$$

while the inverse transformation $C_{T,Tz}^{-1}$ reads

$$C_{T,Tz}^{-1} : \boldsymbol{x} \mapsto \begin{cases} \boldsymbol{x}, & \text{if } T\boldsymbol{z} \in \Omega \\ T\boldsymbol{x}, & \text{if } \boldsymbol{z} \in \Omega \end{cases}. \tag{29}$$

Additionally, one can compute $TC_{T,z}^{-1}$,

$$TC_{T,z}^{-1} : \boldsymbol{x} \mapsto \begin{cases} T\boldsymbol{x}, & \text{if } \boldsymbol{z} \in \Omega \\ TT\boldsymbol{x}, & \text{if } T\boldsymbol{z} \in \Omega \end{cases} = \begin{cases} T\boldsymbol{x}, & \text{if } \boldsymbol{z} \in \Omega \\ \boldsymbol{x}, & \text{if } T\boldsymbol{z} \in \Omega \end{cases} \tag{30}$$

and verify that indeed

$$C_{T,Tz}^{-1}\boldsymbol{x} = TC_{T,z}^{-1}\boldsymbol{x}. \tag{31}$$

Leveraging the identities in Eqs. (28) and (31), one can finally show that the overall map $\widetilde{g}_{\boldsymbol{\theta}}$ is equivariant with respect to the transformation $T$,

$$\widetilde{g}_{\boldsymbol{\theta}}(T\boldsymbol{z}) = C_{T,Tz}^{-1} \, g_{\boldsymbol{\theta}}(C_{T,Tz} \, T\boldsymbol{z}) = TC_{T,z}^{-1} \, g_{\boldsymbol{\theta}}(C_{T,z} \, \boldsymbol{z}) = Tg_{\boldsymbol{\theta}}(\boldsymbol{z}), \tag{32}$$

which proves the initial equivariance condition in Eq. (23).

---

[4]Note that while the subscript $T, z$ means that the forward canonical transformation is applied to the input $\boldsymbol{z}$, the subscript $T, Tz$ means that the transformation is applied to the transformed input $T\boldsymbol{z}$.

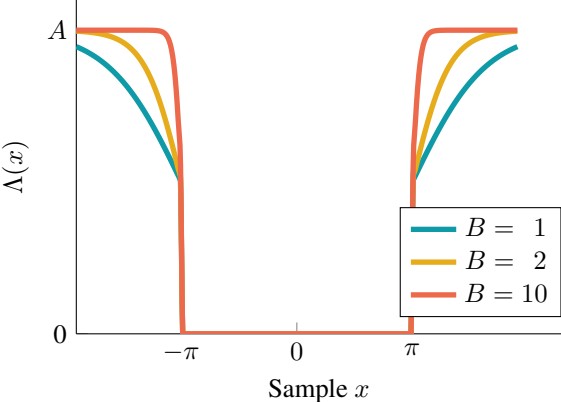

Figure 6: Example of a penalty term with $\lambda(x) = |x| - \pi$. The penalty term is zero for $x \in [-\pi, \pi]$ and approaches $A$ as $x \to \pm\infty$. The parameter $B$ controls the scaling of the penalty gradient.

An essential part of the canonicalization is that the map $g_{\boldsymbol{\theta}}$ *must not* move the canonicalized sample $C_{T,z} z$ *outside* the canonical cell, i.e., into $\mathbb{R}^n \setminus \Omega$. This requirement arises because if the map $g_{\boldsymbol{\theta}}$ maps a sample outside of the canonical cell $\Omega$—that is, if $g_{\boldsymbol{\theta}}(C_{T,z}\, \boldsymbol{z}) \notin \Omega$— then it is possible for two distinct inputs $\boldsymbol{z}_1 \neq \boldsymbol{z}_2$ with $\boldsymbol{z}_1, \boldsymbol{z}_2 \in \mathbb{R}^n$ to be mapped to the same output via canonicalization and transformation: $g_{\boldsymbol{\theta}}(\boldsymbol{z}_1) = g_{\boldsymbol{\theta}}(\boldsymbol{z}_2)$. This leads to a loss of *injectivity* and, consequently, the transformation $g_{\boldsymbol{\theta}}$ is no longer *bijective*. This poses a problem, as NFs require the map $g_{\boldsymbol{\theta}}$ to be bijective in order to perform density estimation via Eq. (2). As described in Sec. 2.3.3, this constraint can be numerically enforced using a penalty term $\Lambda : \boldsymbol{x} \in \mathbb{R}^n \to \Lambda(\boldsymbol{x}) \in \mathbb{R}$, which is zero for $\boldsymbol{x} \in \Omega$ and greater than zero for $\boldsymbol{x} \notin \Omega$. Furthermore, it is essential that the gradient $\partial_{\boldsymbol{z}}\Lambda(g_{\boldsymbol{\theta}}(\boldsymbol{z}))$ points towards the canonical cell $\Omega$. This ensures that if the NF pushes a sample $\widetilde{\boldsymbol{z}} = g_{\boldsymbol{\theta}}(C_{T,z}\, \boldsymbol{z})$ outside of $\Omega$, the gradient of $\Lambda$ acts to pull it back into the cell. Further details on the penalty term and the enforcement of bijectivity are provided in Sec. 2.3.3 and further elaborated in App. D.

## D  PENALTY TERM FOR THE KL DIVERGENCE

In Eq. (10) from Sec. 2.3.3, we introduced a penalty term that is necessary to numerically enforce the bijectivity required for the NF to serve as a valid transport map between probability densities. In this section, we further elaborate on this penalty term and provide an example in Fig. 6.

Crucially, the penalty term $\Lambda(\boldsymbol{x})$ and the associated penalty function $\lambda(\boldsymbol{x})$ are necessary for ensuring that the NF $g_{\boldsymbol{\theta}}$ does not map samples outside of the canonical cell $\Omega$. For convenience, we recall the penalty term,

$$\Lambda(\boldsymbol{x}) = A \cdot \sigma(B \cdot \lambda(\boldsymbol{x})) \cdot \Theta(\lambda(\boldsymbol{x})), \tag{33}$$

where the set $\{A,\, B\}$ denotes all hyperparameters, while $\sigma(\cdot)$ and $\Theta(\cdot)$ refer to the sigmoid and the Heaviside theta functions, respectively.

Fig. 6 shows an example for a penalty term for the canonical cell[5] $\Omega = \{x \in \mathbb{R} : |x| \leq \pi\}$. The function $\lambda(x) = |x| - \pi$ is chosen so that it becomes zero at the boundary $|x| = \pi$ and positive outside the canonical cell, i.e., for $|x| > \pi$. Correspondingly, the penalty term $\Lambda(x)$ is zero for all $x \in [-\pi, \pi]$ and smoothly approaches the value $A$ as $x \to \pm\infty$. The parameter $B$ controls the scaling of the gradient of the penalty term.

## E  GENERALIZATION OF SESAMO

### E.1  $\mathbb{Z}_M$ STOCHASTIC MODULATION

The stochastic modulation for the $\mathbb{Z}_2$ symmetry introduced in Sec. 3.1 can be generalized to a $\mathbb{Z}_M$ symmetry. The transformation $S_u$ randomly rotates a two-dimensional vector $\boldsymbol{x} \equiv (x_1, x_2)^T \in \mathbb{R}^2$

---

[5]Note that the example is in one-dimensional space $\mathbb{R}$ but can be straightforwardly generalized.

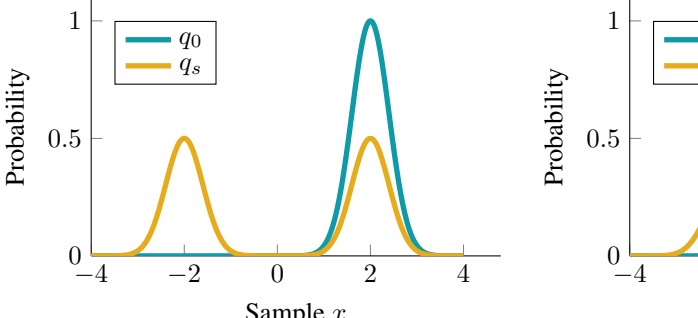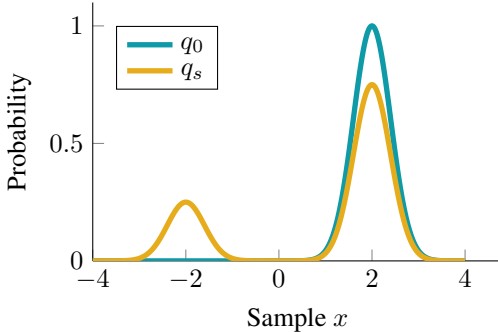

Figure 7: Prior Gaussian distribution $q_0$ with mean $\mu = 2$ and standard deviation $\sigma = 1$. The transformation $S_u$, implementing the $\mathbb{Z}_2$ symmetry, randomly flips the sign of a sample $x_i \sim q_0$ with a probability determined by the breaking parameter $b$. When $b = \ln 0.5$ (left), the resulting distribution $q_s$ (yellow) is symmetric around zero, with both modes carrying equal probability mass. When $b = \ln 0.25$ (right), according to Eq. (38), the sign flip occurs with probability $p_{S,b} = 0.25$, leading to asymmetric modes at $\mu = \pm 2$ that carry 25% and 75% of the total probability mass, respectively.

| $b$ | $p_{S,b}(u=0)$ | $p_{S,b}(u=1)$ |
|---|---|---|
| $0$ | $0$ | $1$ |
| $\ln 0.5$ | $1/2$ | $1/2$ |
| $-\infty$ | $1$ | $0$ |

Table 2: Probability $p_{S,b}$ of *not* flipping ($u = 0$) and flipping ($u = 1$) the sign of the input $x$ for examples of the breaking parameter $b$, including the even case and the edge cases.

about the origin by an angle of $2\pi u/M$, i.e.,

$$S_u : \boldsymbol{x} \to \begin{pmatrix} \cos \frac{2\pi u}{M} & -\sin \frac{2\pi u}{M} \\ \sin \frac{2\pi u}{M} & \cos \frac{2\pi u}{M} \end{pmatrix} \boldsymbol{x} \qquad \text{with} \qquad u \sim \mathcal{U}_{\text{disc}}(0, M), \qquad (34)$$

where $u \sim \mathcal{U}_{\text{disc}}(0, M)$ is a discrete uniform random variable taking values in the set $\{0, 1, 2, \dots, M-1\}$. The modulation probability is therefore given by $p_S = 1/M$. To ensure the bijectivity of the transformation $S_u$, the penalty term $\widetilde{\Lambda}$ is added to the KL divergence in Eq. (10), where

$$\widetilde{\Lambda}(\boldsymbol{x}) = \Lambda[\lambda_-(\boldsymbol{x})] + \Lambda[\lambda_+(\boldsymbol{x})], \qquad (35)$$

and the bijectivity function is expressed as

$$\lambda_\pm(\boldsymbol{x}) = -\tan(\pi/M)\, x_1 \pm \frac{x_2}{(1 + \tan(\pi/M))^2}. \qquad (36)$$

The canonical cell defined by this penalty term corresponds to a sector of angular width $2\pi/M$ centered around the $x_1$-axis, with boundaries at angles $\pm\pi/M$. The bijectivity function then measures the distance of a sample to the border of the canonical cell. For more details on the penalty term, we refer back to Sec. 2.3.3.

### E.2 Broken $\mathbb{Z}_2$ Stochastic Modulation

In the main text, the *exact* $\mathbb{Z}_2$ symmetry was considered to illustrate how canonicalization and SESaMo transform the base density. A $\mathbb{Z}_2$ symmetry is called *exact* when both modes (as shown in Fig. 1 and Fig. 2) carry equal probability mass. In the following, we extend this to a more general case where the probability mass is unevenly distributed across the modes.

It is important to note that under these conditions, the canonicalization approach faces challenges. Specifically, it is no longer sufficient to learn a single mode and evenly distribute the probability mass

among the others. In contrast, SESaMo, owing to its greater flexibility, can effectively handle this asymmetry. To accommodate such cases, a learnable *breaking parameter* $b \in \mathbb{R}^-$ is introduced to account for the imbalance in probability mass between the modes. When $b \to 0$, the sign of $\boldsymbol{x}$ is always flipped, whereas in the limit $b \to -\infty$, the sign is never flipped. The transformation $S_u$ for a broken $\mathbb{Z}_2$ symmetry therefore yields

$$S_u : \boldsymbol{x} \to \begin{cases} \boldsymbol{x} & \text{if } u = 0 \\ -\boldsymbol{x} & \text{if } u = 1 \end{cases} \qquad \text{with} \qquad u \sim \mathcal{B}(e^b) \qquad \text{and} \qquad b \in \mathbb{R}^- \,, \qquad (37)$$

where $\mathcal{B}(e^b)$ denotes a Bernoulli distribution. Note that when $b = \ln 0.5$, the transformation reduces to the symmetric $\mathbb{Z}_2$ case, where each mode is selected with equal probability. Tab. 2 shows the modulation probability $p_{S,b}$ for the even case and the edge cases of the breaking parameter $b$ discussed above. For an arbitrary breaking parameter $b$, the modulation probability $p_{S,b}$ is given by

$$p_{S,b} = \begin{cases} 1 - e^b & \text{if } u = 0 \\ e^b & \text{if } u = 1 \,, \end{cases} \qquad (38)$$

where $u \sim \mathcal{B}(e^b)$. The corresponding bijectivity constraint, used in the penalty term $\Lambda$ introduced in Eq. (10), reads

$$\lambda(\boldsymbol{x}) = -\sum_{i=1}^{N} x_i \,, \qquad (39)$$

where the sum is taken over of all components of the vector $\boldsymbol{x} \in \mathbb{R}^N$. The breaking parameter $b$ is used in the exponential to ensure numerically stable simulations, which becomes particularly important in the limits $p_{S,b} \to 0$ and $p_{S,b} \to 1$.

Fig. 7 (left) shows a one-dimensional Gaussian distribution $q_0$ (blue), centered at $x = 2$ with standard deviation $\sigma = 1$. Applying the stochastic modulation $S_u$ corresponding to the $\mathbb{Z}_2$ symmetry, with the breaking parameter $b = \ln 0.5$, yields a new distribution $q_s$ (yellow) that is symmetric around zero. In this case, the probability mass is equally distributed across both modes. When the breaking parameter $b \neq \ln 0.5$, the stochastic modulation accounts for the imbalance between the modes, resulting in unequal probability masses in the transformed density $q_s$. Fig. 7 (right) shows an example for $b = \ln 0.25$, where the mode at $x < 0$ carries less mass than the one at $x > 0$.

Numerically, a so-called *breaking ratio* can be estimated by counting the number of samples in each mode of the distribution:

$$\hat{R} = \frac{N_+ - N_-}{N_+ + N_-} = 1 - 2e^b \,, \qquad (40)$$

where $N_+$ and $N_-$ denote the number of samples in the positive and negative modes of $q_s$, respectively. As an example, the experiments for the Hubbard model presented in the main text feature a broken $\mathbb{Z}_4$ symmetry, composed of an exact $\mathbb{Z}_2$ and a broken $\mathbb{Z}_2$ symmetry. SESaMo is able to learn this *broken* $\mathbb{Z}_4$ symmetry by combining an exact and a broken $\mathbb{Z}_2$ transformation, i.e., effectively modulating the sign of one of two field components.

### E.3 Broken $(\mathbb{Z}_2)^{N_x}$ Stochastic Modulation

In the main text, we presented results for the Hubbard model for both small and large spatial lattice extents, i.e., $N_x = 2$ and $N_x = 18$. The Hubbard models exhibits a combination of $N_x$ distinct $\mathbb{Z}_2$ symmetries, forming a $(\mathbb{Z}_2)^{N_x}$ symmetry. In terms of the stochastic modulation, we introduce breaking parameters $b \in \mathbb{R}^n$ with $n \equiv 2^{N_x}$. The stochastic variable $u$ is drawn from a categorical distribution $\mathrm{Cat}(b)$ with log-odds $b$ and $u = 0, 1, \ldots, 2^{N_x} - 1$. The canonical cell is defined as $\Omega = \{\boldsymbol{x} \in \mathbb{R}^{N_x \times N_t} \mid \sum_t x_{it} > 0 \;\; \forall i = 0, 1, \ldots, N_x - 1\}$ and the stochastic modulation is given by

$$S_u : \boldsymbol{x} \to \begin{cases} (\phantom{-}1, \ldots, \phantom{-}1, \phantom{-}1, \phantom{-}1)^T \odot \boldsymbol{x} & \text{if } u = 0 \\ (\phantom{-}1, \ldots, \phantom{-}1, \phantom{-}1, -1)^T \odot \boldsymbol{x} & \text{if } u = 1 \\ (\phantom{-}1, \ldots, \phantom{-}1, -1, \phantom{-}1)^T \odot \boldsymbol{x} & \text{if } u = 2 \\ (\phantom{-}1, \ldots, \phantom{-}1, -1, -1)^T \odot \boldsymbol{x} & \text{if } u = 3 \\ \quad \vdots \\ (-1, \ldots, -1, -1, -1)^T \odot \boldsymbol{x} & \text{if } u = 2^{N_x} - 1 \,, \end{cases} \qquad \text{with} \qquad u \sim \mathrm{Cat}(\mathrm{b}) \,, \qquad (41)$$

where $\odot$ denotes component-wise multiplication. This formulation ensures that $\boldsymbol{x}$ can be transformed to one of the $2^{N_x}$ orthans.

We can combine this broken symmetry with the exact $\mathbb{Z}_2$ symmetry present in the system, resulting in an effective broken $(\mathbb{Z}_2)^{N_x-1} \otimes$ exact $\mathbb{Z}_2$ symmetry. For $N_x = 2$, this reduces to a broken $\mathbb{Z}_4$ symmetry.

## F  STOCHASTIC MODULATION FOR CONTINUOUS SYMMETRIES

In Sec. 3.1, the stochastic modulation $S_u$ was introduced for discrete symmetries, where $S_u$ has a finite number of possible outcomes, each selected according to the modulation probability $p_{S,b}$. This approach is well-suited for discrete symmetries such as sign-flip or $\mathbb{Z}_M$ symmetries. However, it is not applicable to continuous symmetries—such as rotational or translational symmetries—where the transformation space is uncountably infinite. In these cases, a modified formulation of stochastic modulation is required to account for the continuous nature of the symmetry group.

The continuous stochastic modulation proceeds as follows: first, draw a sample $u$ from a distribution $q_u(u)$, which can be the uniform distribution $\mathcal{U}(0, 1)$. Then, apply a trainable map $h_b : [0, 1) \to [0, 1)$ with parameters $b$ to obtain $h_b(u)$. This output parametrizes a continuous transformation $R_{h_b(u)}$, such as a rotation matrix where the rotation angle is determined by $h_b(u)$. The stochastic transformation is thus given by

$$S_u : \boldsymbol{x} \to R_{h_b(u)}\boldsymbol{x}\,. \tag{42}$$

The modulation probability, which enters the density transformation in Eq. (15), follows from the change-of-variable formula of the transformation $R_{h_b(u)}$ and can be expressed as

$$p_{S,b}(u) = q_u(u) \cdot \left| \det\left( \frac{\partial R_{h_b(u)}^{-1}}{\partial u} \right) \right|\,, \tag{43}$$

where $q_u(u)$ is the probability density of $u$ and the determinant captures the local volume change under the inverse transformation $R_{h_b(u)}^{-1}$.

### F.1  BROKEN AND EXACT $U(1)$ STOCHASTIC MODULATION

In Sec. 4.2, the complex $\phi^4$ scalar field theory is introduced, in which the action $f[\boldsymbol{x}]$ (as defined in Eq. (21)) remains invariant under a $U(1)$ transformation of the form

$$R_\varphi = e^{2\pi i \varphi}, \tag{44}$$

where the angle $\varphi$ lies in the interval $[0, 1)$. If a term $\alpha \mathrm{Re}[\boldsymbol{x}]$ is added to the action $f[\boldsymbol{x}]$, this $U(1)$ symmetry is broken, meaning that the Boltzmann-like density $p(\boldsymbol{x}) = \exp\left(-f[\boldsymbol{x}]\right)/Z$ becomes dependent on the angle $\varphi$. This angular dependence can be captured within the stochastic modulation framework by introducing a trainable map $\varphi \equiv h_b(u)$. In particular, a spline flow (73) is used for this purpose. The modulation probability in Eq. (43) then simplifies to

$$p_{S,b}(u) = \frac{1}{2\pi} \left| \det\left( \frac{\partial h_b(u)}{\partial u} \right) \right|^{-1}\,, \tag{45}$$

where the chain rule is used to compute $\partial R_{h_b(u)}^{-1}/\partial u$ in Eq. (43), as well as the fact that

$$\left| \det\left( \frac{\partial R_{h_b(u)}^{-1}}{\partial h_b} \right) \right| = \frac{1}{2\pi}\,. \tag{46}$$

This is given because the rotation $R_\varphi = e^{2\pi i \varphi}$ in Eq. (44) corresponds to a full angular cycle over the interval $[0, 1)$, scaling the Jacobian by the full rotation angle $2\pi$. Meanwhile, we used $q_u = 1$ since $u$ is sampled from a uniform distribution on $[0, 1)$, which has a constant density of one.

The sample $\boldsymbol{x}$ must be completely real before applying the stochastic modulation. This means that a prior sample $\boldsymbol{z} = \boldsymbol{z}_1 + i\boldsymbol{z}_2$, where $\boldsymbol{z}_1, \boldsymbol{z}_2 \in \mathbb{R}^N$, must satisfy $\boldsymbol{z}_2 = 0$, i.e., it lies on the real axis,

and is transformed by an NF $g_\theta : \mathbb{R}^N \to \mathbb{R}^N$. After applying the stochastic modulation $R_{h_b(u)}$, the sample $\boldsymbol{x}$ becomes complex-valued, given by

$$\boldsymbol{x} = e^{2\pi i h_b(u)} g_\theta(\boldsymbol{z}_1) \,. \tag{47}$$

Note that by omitting the spline flow $h_b$ and using $h \equiv \mathbb{1}$, an exact $U(1)$ symmetry can be enforced instead of a broken one. Furthermore, this approach can similarly be used to enforce a broken or exact rotational $SO(2)$ symmetry.

# G    TECHNICAL DETAILS OF THE PHYSICAL THEORIES

In this section, we discuss some fundamental aspects of the complex $\phi^4$ theory and the Hubbard model that are relevant to our study.

## G.1    THE COMPLEX $\phi^4$ SCALAR FIELD THEORY IN TWO DIMENSIONS

In recent years, the $\phi^4$ theory has become a popular benchmark for generative models in the machine learning community (66; 67; 68). Originally developed as a physical model, it describes interacting particles with integer spin. On a finite lattice with points $j \in V$, the theory is specified by the action

$$\widetilde{f}[\boldsymbol{\varphi}] = \sum_{j \in V} \left[ \frac{a^2}{2} \sum_{\hat{\mu}=1}^{2} \frac{\left(\boldsymbol{\varphi}_{j+a\hat{\mu}} - \boldsymbol{\varphi}_j\right)^2}{a^2} + \frac{m_0^2}{2}\boldsymbol{\varphi}_j^2 + \frac{g_0}{4!}\boldsymbol{\varphi}_j^4 \right] \,, \tag{48}$$

where $\boldsymbol{\varphi}_j$ denotes the field value at site $j$. The first term inside the brackets corresponds to the kinetic term, the second is the mass term governed by the bare mass $m_0$, and the quartic $\varphi^4$ term describes the interaction, weighted by the bare coupling strength $g_0$. Using the more standard redefinitions (similarly adopted by Nicoli et al. (14))

$$\boldsymbol{\varphi} = (2\kappa)^{1/2}\,\boldsymbol{x} \,, \qquad (am_0)^2 = \frac{1-2\lambda}{\kappa} - 4 \,, \qquad a^2 g_0 = \frac{6\lambda}{\kappa^2} \,, \tag{49}$$

we rewrite the action in the form presented in the main text:

$$f[\boldsymbol{x}] = \sum_{j \in V} \left[ -2\kappa \sum_{\hat{\mu}=1}^{2} (\boldsymbol{x}_j \boldsymbol{x}_{j+\hat{\mu}}) + (1-2\lambda)\boldsymbol{x}_j^2 + \lambda \boldsymbol{x}_j^4 + \alpha \mathrm{Re}[\boldsymbol{x}_j] \right] \,. \tag{50}$$

Here, $\lambda$ is known as the coupling parameter, while $\kappa$ is the hopping parameter. Additionally, we added a term $\alpha \mathrm{Re}[\boldsymbol{x}_j]$ to progressively break the $U(1)$ symmetry of the $\phi^4$ theory as the parameter $\alpha$ increases. Such a symmetry-breaking term also arises in quantum field theories with non-degenerate particle flavor masses, providing a physically motivated example.

## G.2    THE HUBBARD MODEL IN TWO DIMENSIONS

The Hubbard model is a fundamental model in condensed matter physics that describes how electrons interact on a fixed lattice of ions (70). By neglecting lattice vibrations and other atomic excitations, it captures the essential physics of electrons hopping between valence orbitals and interacting through their electric charge. This is further illustrated in Fig. 8. We describe the system in the so-called *spin basis*, where the degrees of freedom correspond to spin-up and spin-down electrons. Other basis choices exist but are not considered here.

The action of the system is given by (71)

$$f[\boldsymbol{x}] = \frac{1}{2\widetilde{U}} \sum_{j,k \in V} \boldsymbol{x}_{jk}^2 - \log \det M\,[\boldsymbol{x}] - \log \det M[-\boldsymbol{x}] \,, \tag{51}$$

where $\widetilde{U}$ denotes the on-site Coulomb-like interaction strength, $\boldsymbol{x}$ are auxiliary bosonic fields, and the subscripts $j, k$ label the spatial and temporal lattice sites in the lattice volume $V$, respectively. Since we do not consider a temporal extent throughout this manuscript, i.e. $N_t = 1$, we have dropped the index $k$ in Sec. 4.2 for brevity. Lastly, the fermion matrix $M$ is defined as

$$M\,[x]_{j'k',jk} = \delta_{j',k}\delta_{j',k} - [e^h]_{j',k}e^{\phi_{jk}}\mathcal{B}_{k'}\delta_{k',k+1} \,. \tag{52}$$

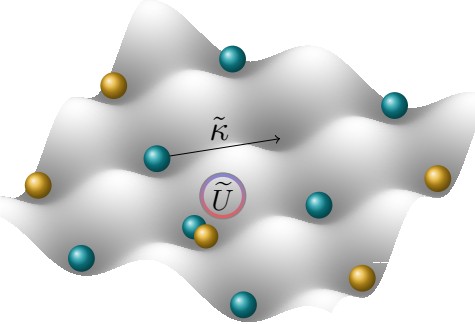

Figure 8: Illustration of a lattice described by the Hubbard model. Blue and red circles represent spin-up and spin-down electrons, respectively. The hopping term $\tilde{\kappa}$ allows electrons to move between neighbouring lattice sites, while the on-site Coulomb interaction $\widetilde{U}$ penalizes the presence of two electrons with opposite spins at the same site.

Here, $h = \tilde{\kappa}\delta_{\langle j',j\rangle}$ is the hopping matrix, where $\tilde{\kappa}$ is the hopping amplitude and $\delta_{\langle j',j\rangle}$ enforces hopping only between nearest neighbours $j', j$ on the lattice, and $\mathcal{B}_t$ is a factor implementing periodic (anti-periodic) boundary conditions in the temporal direction for $N_t = 1$ ($N_t > 1$). The action in Eq. (51) consists of two main contributions: the Gaussian term, which encodes the on-site interaction, and the fermionic term, represented by the product of fermion matrices, which captures the electron hopping dynamics across the lattice.

The Boltzmann-like density of the Hubbard model features widely separated modes, which can lead to ergodicity problems and biased estimates of observables when using Monte Carlo-based sampling methods such as Hybrid Monte Carlo (HMC) (74). NFs have demonstrated the ability to overcome these challenges, particularly when they incorporate prior knowledge of the system's symmetries (38).

## H    ADDITIONAL NUMERICAL EXPERIMENTS

In this section, we present additional experiments for the Gaussian mixture model, the Hubbard model, and the $\phi^4$ theory.

### H.1    GAUSSIAN MIXTURE

The Gaussian mixture model introduced in Sec. 4 exhibits a multi-modal density, where locating all modes is poses a significant challenge for RealNVP. This issue is mitigated by applying canonicalization and further improved with SESaMo, which achieves higher accuracy. Fig. 9 (left) shows the ESS as a function of GPU training time in minutes. The solid lines and shaded regions indicate the mean and standard deviation over ten models trained with different seeds. Both canonicalization and SESaMo lead to faster convergence compared to RealNVP, which suffers from strong fluctuations due to frequent mode collapse.

### H.2    THE HUBBARD MODEL IN TWO DIMENSIONS

In Fig. 9 (right), the ESS is shown as a function of the GPU training time for the Hubbard model. The solid lines and shaded regions indicate the mean and standard deviation over ten models trained with different seeds. SESaMo not only achieves higher accuracy than both canonicalization and RealNVP, but also converges faster than RealNVP. The canonicalization method fails to capture the unequal probability masses across the modes, as illustrated in Fig. 5, while RealNVP suffers from mode-dropping. In contrast, SESaMo successfully identifies all four modes and accurately predicts their relative probabilities.

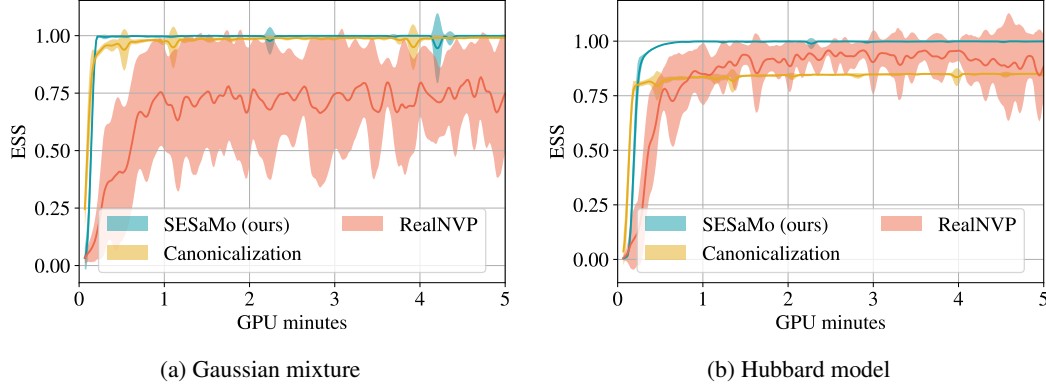

(a) Gaussian mixture            (b) Hubbard model

Figure 9: ESS as a function of the GPU training time (minutes) for the Gaussian mixture (left) and the Hubbard model (right). Solid lines represent the mean and shaded areas indicate the standard deviation across ten models trained with different seeds. The results show that SESaMo achieves a higher ESS compared to both canonicalization and RealNVP.

The effect of the broken $\mathbb{Z}_2$ symmetry becomes more pronounced as the inverse temperature $\beta$ increases. To investigate this behaviour, we train SESaMo and canonicalization models for values of $\beta \in [1,4]$, as shown in Fig. 10 (left). SESaMo consistently achieves high accuracy across all values of $\beta$, while the canonicalization method exhibits significantly lower accuracy. This demonstrates that SESaMo successfully learns the broken $\mathbb{Z}_2$ symmetry.

To further verify whether the probability is predicted correctly, we compare against the ground truth. In Fig. 10 (right), the breaking ratio $R$ from Eq. (40) is shown, where $N_\pm$ can be computed analytically by integrating the probability distribution $p(\boldsymbol{x})$ for a volume $V = 2 \times 1$, i.e., $\boldsymbol{x} = (x_1, x_2) \in \mathbb{R}^2$. The probability distribution[6] is known up to a constant factor and given by

$$p(\boldsymbol{x}) \propto h(\boldsymbol{x})h(-\boldsymbol{x})e^{-\frac{x_1^2+x_2^2}{U\beta}} , \tag{53}$$

where

$$h(\boldsymbol{x}) = \cosh\left(\frac{x_1 + x_2}{2}\right) + \cosh\left(\frac{x_1 - x_2}{2}\right)\cosh(\tilde{\kappa}) . \tag{54}$$

The theoretical prediction of the breaking ratio $R$ matches perfectly with the expression $R = 1 - 2e^b$ obtained from the learned breaking parameter $b$.

## H.3   THE REAL $\phi^4$ SCALAR FIELD THEORY IN TWO DIMENSIONS

In Sec. 4.2 and G.1, we introduced the *complex* $\phi^4$ scalar field theory in two dimensions. In its general form, this theory consists of complex-valued fields.

Most recent works in the context of generative models (see, e.g., (13; 14)), however, have focused on *real* scalar fields. Under this assumption, the $\phi^4$ theory belongs to the same universality class as the Ising model and serves as an instructive toy model for exploring spontaneous symmetry breaking and the Higgs mechanism (65). Assuming *real* scalar fields, the action in Eq. (21) simplifies to

$$f[\boldsymbol{x}] = \sum_{j \in V}\left[-2\kappa\sum_{\hat{\mu}=1}^{2}(\boldsymbol{x}_j\boldsymbol{x}_{j+\hat{\mu}}) + (1 - 2\lambda)\boldsymbol{x}_j^2 + \lambda\boldsymbol{x}_j^4 + \alpha\boldsymbol{x}_j\right] , \tag{55}$$

with $\boldsymbol{x} \in \mathbb{R}^n$. This form of the action corresponds to the one studied in Ref. (14; 72), up to the addition of a symmetry-breaking factor $\alpha\boldsymbol{x}$. The coefficient $\alpha$ introduces an exponential suppression of the probability with respect to the field $\boldsymbol{x}$, thereby explicitly breaking the $\mathbb{Z}_2$ symmetry when $\alpha > 0$. In this context, the symmetry-breaking parameter $b$ introduced in App. E can be learned such

---

[6]Note that this distribution is exact for $V = 2 \times 1$. For larger volumes, it becomes exact only in the strong-coupling limit $U \to \infty$ while keeping $\beta$ fixed.

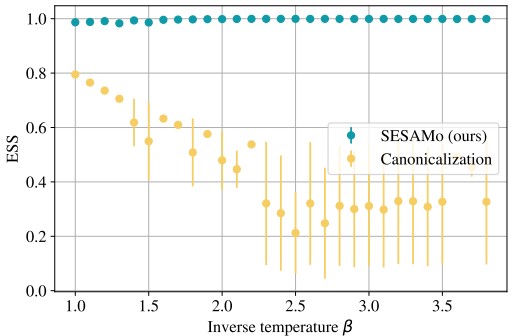 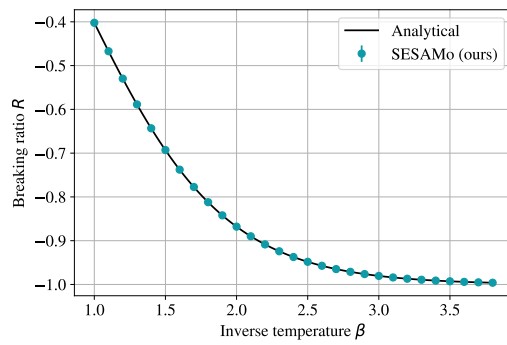

(a) ESS as a function of the inverse temperature $\beta$.  (b) $R$ as a function of the inverse temperature $\beta$.

Figure 10: **Left**: ESS for different values of the inverse temperature $\beta$. The blue and yellow markers correspond to canonicalization and SESaMo, respectively. Means and standard deviations are computed by averaging over three independently trained models (for each method) using three different random seeds. **Right**: Breaking ratio $R$ as a function of $\beta$. The analytical curve (yellow) is obtained by integrating the analytically derived probability weight (see Eq. (79) in Ref. (74)). The numerical estimate from Eq. (40), computed using a trained SESaMo model, agrees with the analytical result within error bars. The uncertainties—often too small to be visible at the scale of the plot—are estimated by averaging over three independently trained models with different seeds.

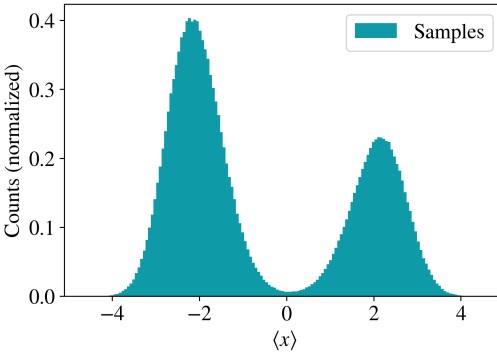 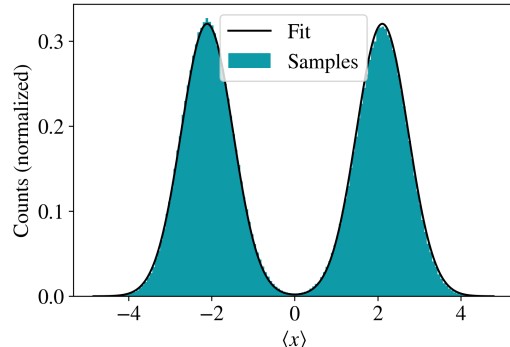

(a) Magnetization for broken $\mathbb{Z}_2$ ($\alpha = 0.001$).  (b) Magnetization for unbroken $\mathbb{Z}_2$ ($\alpha = 0$).

Figure 11: Histograms of the magnetization for real $\phi^4$ scalar field theory for a broken $\mathbb{Z}_2$ symmetry (left, $\alpha = 0.001$) and an exact $\mathbb{Z}_2$ symmetry (right, $\alpha = 0$). Samples for the histograms are drawn from two SESaMo models trained for the corresponding values of the breaking factor $\alpha$.

that SESaMo redistributes the probability mass of the learned probability in accordance with the asymmetry of the target distribution. We train SESaMo using the REINFORCE estimator of the KL divergence discussed in Sec. 3.2. Additional hyperparameters and experimental details are provided in App. I. Unless stated otherwise, all experiments in this setting are conducted on lattices of size $16 \times 8$, with action parameters fixed to $\kappa = 0.3$ and $\lambda = 0.022$.

Since the theory now consists of scalar real fields in two dimensions, it enters the so-called broken phase for couplings $\{\kappa, \lambda\} = \{\geq 0.3, 0.022\}$. This phase is characterized by a bimodal probability density with the centers of the modes located at the vacuum expectation values (VEVs) (72) of the theory. When $\alpha = 0$, both modes are identical, and the resulting distribution is symmetric. In this case, both SESaMo and canonicalization are able to accurately learn the target distribution, achieving high ESS without mode collapse (72). In the following, we compare the performance of SESaMo and canonicalization in the case $\alpha > 0$, where the $\mathbb{Z}_2$ symmetry of the double-well potential is explicitly broken. To study this scenario, we trained different models using both approaches for increasing values of $\alpha$. The results are shown in Fig. 12 (left), which displays the ESS obtained from models trained for a $\phi^4$-theory defined on a lattice of size $V = 16 \times 8$ for various values of $\alpha$.

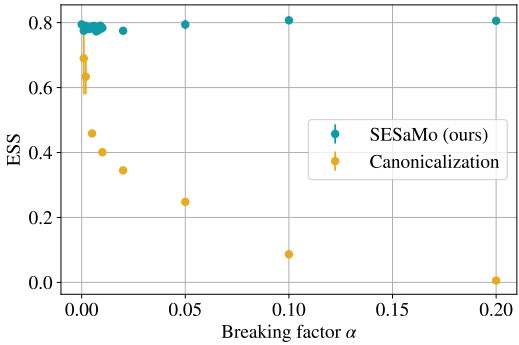 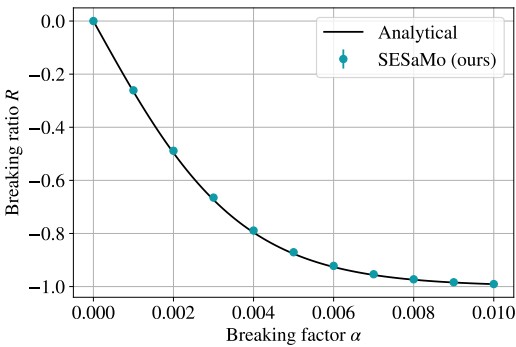

(a) ESS as a function of the breaking factor $\alpha$.     (b) $R(\alpha)$ as a function of the breaking factor $\alpha$.

Figure 12: **Left**: ESS for different values of the breaking factor $\alpha$. The blue and yellow markers refer to canonicalization and SESaMo, respectively. Mean and standard deviations are computed by averaging three models (for both approaches) trained with three different seeds. **Right**: Breaking ratio $R$ for different values $\alpha$. The analytical (yellow) curve is obtained by plotting Eq. (58) as a function of $\alpha$. The numerical estimate in Eq. (40), obtained with a trained SESaMo model, is compatible with the analytical result within errors. The uncertainties (sometimes too small to be visible in the scale of the plot) are estimated by averaging three models trained with three different seeds.

Yellow and blue markers indicate results from SESaMo and canonicalization, respectively. Error bars represent standard deviations computed from three independently trained models with different seeds. Crucially, while the performance of SESaMo and canonicalization is comparable at $\alpha = 0$, the ESS of canonicalization drops to zero as $\alpha$ increases, and the potential becomes increasingly asymmetric. In contrast, SESaMo maintains a stable ESS across the entire range of $\alpha$, thanks to the stochastic modulation enabled by the learned symmetry-breaking parameter.

Interestingly, this analysis can be made fully quantitative. The distribution of the magnetization for the $\phi^4$ theory (see Fig. 11) yields a Gaussian distribution with two modes located at the VEVs$= \pm\mu$, and is modulated by the symmetry-breaking factor $\alpha$,

$$\tilde{f}(x) = A \left( e^{-\frac{(x-\mu)^2}{2\sigma^2}} + e^{-\frac{(x+\mu)^2}{2\sigma^2}} \right) \cdot e^{-\alpha V x}, \tag{56}$$

where $V = 16 \times 8$ is the volume of the lattice. The parameters $\{A, \sigma, \mu\}$ can be inferred from a numerical fit of the histogram at $\alpha = 0$ (see Fig. 11b), yielding

$$A = 0.499(2), \qquad\qquad \mu = 2.126(3), \qquad\qquad \sigma = 0.629(3).$$

These parameters fully characterize the distribution defined in in Eq. (55). To quantify the effect of symmetry breaking, we define $N_+(\alpha)$ and $N_-(\alpha)$ as the integrated probability mass over the right and left modes, respectively:

$$N_-(\alpha) = \int_{-\infty}^{0} dx \, \tilde{f}_\alpha(x) \qquad\qquad \text{and} \qquad\qquad N_+(\alpha) = \int_{0}^{\infty} dx \, \tilde{f}_\alpha(x). \tag{57}$$

We then define the breaking ratio $R$ as the relative imbalance between the two modes $N_+$ and $N_-$. Using standard Gaussian integrals, this ratio can be computed analytically, resulting in

$$R(\alpha) \equiv \frac{N_+(\alpha) - N_-(\alpha)}{N_+(\alpha) + N_-(\alpha)} = 1 - \frac{e^{-V\alpha\mu} \left[ 1 + \mathrm{erf}\left( \tau_-(\alpha) \right) \right] + e^{V\alpha\mu} \left[ 1 + \mathrm{erf}\left( \tau_+(\alpha) \right) \right]}{2 \cosh(V\alpha\mu)} \tag{58}$$

where $\tau_\pm(\alpha)$ are defined by

$$\tau_\pm(\alpha) = \frac{\sigma}{\sqrt{2}} \left( V\alpha \pm \frac{\mu}{\sigma^2} \right). \tag{59}$$

The analytical result from Eq. (58) can be compared to the numerical estimate from Eq. (40). Fig. 10 (right) shows both the analytical prediction and the numerical estimate for the ratio $R(\alpha)$, for breaking factors $\alpha \in [0, 0.01]$. The theoretical value in Eq. (58) and the numerical estimate in Eq. (40), for different $\alpha$, are represented with a solid (black) line and (blue) markers, respectively. For $\alpha = 0$, the

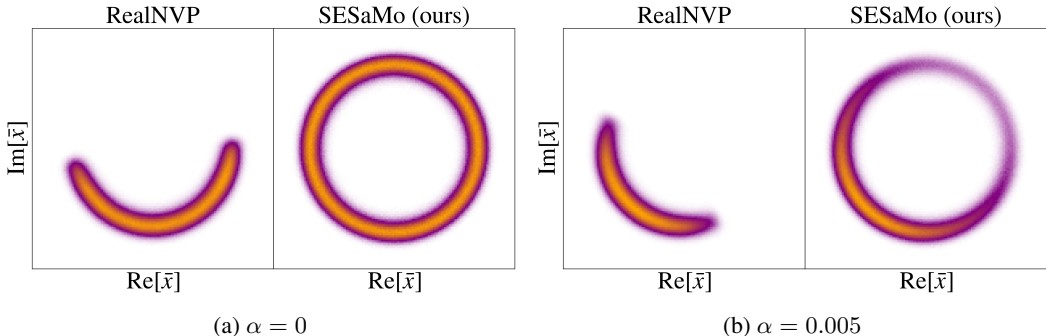

(a) $\alpha = 0$                    (b) $\alpha = 0.005$

Figure 13: **Continuous Symmetries:** Density plot for real and imaginary components of the complex-valued fields of complex $\phi^4$ scalar field theory, as introduced in Sec. 4, and sampled from trained generative models, i.e., RealNVP and SESaMo. The models have been trained to sample from the target density in Eq. (21) for lattices of volume $V = 8 \times 8$ and coupling values $\{\kappa, \lambda\} = \{0.3, 0.022\}$. The models are trained until convergence and the density plots are made by drawing 5 M samples. The left and right plots refer to continuos $U(1)$ symmetries in the unbroken ($\alpha = 0$) and broken ($\alpha = 0.005$) case, respectively. Note that canonicalization is not shown as that approach is not capable of handling continuos symmetries.

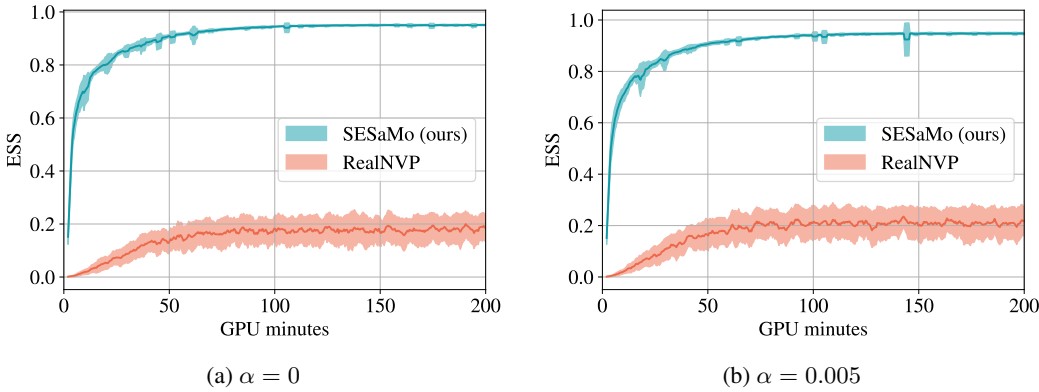

(a) $\alpha = 0$                    (b) $\alpha = 0.005$

Figure 14: ESS as a function of the GPU training time (minutes) for the $\phi^4$ theory experiments (see Fig. 13). The solid line and the shadows represent the mean and the standard deviations of ten models trained with different seeds. The curve shows the substantially faster convergence of SESaMo compared to naive RealNVP. Again, canonicalization is not shown as it cannot straightforwardly incorporate continuos symmetries into the model.

estimated ration from the model is zero, suggesting that the fact the $\mathbb{Z}_2$ symmetry is not broken has been correctly learned by SESaMo. By increasing $\alpha$ the ratio $R$ converges to -1, which corresponds to a fully broken $\mathbb{Z}_2$ symmetry i.e., that is, the probability for $x > 0$ is zero. Crucially, SESaMo is able to always learn the correct *breaking parameter*, hence estimating the correct *breaking ratio $R$* for a broken $\mathbb{Z}_2$-symmetric action.

With this simple example, we conclude that SESaMo is capable of incorporating symmetries inside a flow-based generative model even when those are *broken*. One could foresee the power of this approach in incorporating other types of broken symmetries, such as the chiral symmetry breaking in quantum chromodynamics (QCD) (75). The QCD Lagrangian with two flavors, i.e., up and down quarks, has a broken chiral symmetry due to the different masses of the up and down quarks. Results for a toy model of such scenario were presented in the main text (see Tab. 1) and we further elaborate on them in App. H.4 below.

| Experiment | $N_C$ | $N_L$ | $N_N$ | Activation | $N_B$ | LR | Steps | $\mu$ | Var |
|---|---|---|---|---|---|---|---|---|---|
| GMM | 6 | 4 | 40 | ReLU (Tanh[7]) | 8 k | $5 \times 10^{-4}$ | 10 k | 0 | $1\,(20^8)$ |
| Complex $\phi^4$ | 6 | 4 | 100 | ReLU | 8 k | $5 \times 10^{-4}$ | 400 k | 0 | 1 |
| Hubbard $2 \times 1$ | 6 | 4 | 40 | ReLU (Tanh[8]) | 8 k | $5 \times 10^{-4}$ | 6 k | 0 | 18 |
| Hubbard $18 \times 100$ | 10 | 4 | 100 | ReLU (Tanh[8]) | 8 k | $5 \times 10^{-4}$ | 20 k | 0 | 0.18 |

Table 3: Hyperparameters for the Gaussian mixture model (GMM), the complex $\phi^4$ theory and the Hubbard model. Shown are the number of couplings $N_C$, number of layers $N_L$, number of neurons per layer $N_N$, activation function, batch size $N_B$, learning rate (LR), training steps / epochs, and the mean $\mu$ and variance of the prior Gaussian distribution.

## H.4  THE COMPLEX $\phi^4$ SCALAR THEORY IN TWO DIMENSIONS

In light of these considerations, in the main text (see Tab. 1) we tested how SESaMo is capable of dealing with continuous (broken and unbroken) symmetries, and we showed a remarkable outperformance compared to a naive RealNVP model. Furthermore, in App. F we discussed the details of SESaMo when dealing with continuos symmetries. In this section we complement the results from the main text with some further insights. First, Fig. 13 shows the density of the real and imaginary components of the complex fields $x \in \mathbb{C}$ summed across the lattice volume, i.e., $\text{Re}[\tilde{x}] = \sum_{i \in V} \text{Re}[\tilde{x}_i]$ and $\text{Im}[\tilde{x}] = \sum_{i \in V} \text{Im}[\tilde{x}_i]$. Fig. 13a shows a ring-shaped potential projected on the complex plane stemming from the spontaneous symmetry breaking of an exact $U(1)$ symmetry in the $\phi^4$ theory, which leads to the emergence of Goldstone Bosons (see (76) and Fig. 1 therein). When the $U(1)$ symmetry itself is broken ($\alpha = 0.005$), the probability density around the ring is no more evenly distributed, as it is visualized in the density learned by SESaMo in Fig. 13b. The reader should note that crucially, in the setting of continuous symmetries, only naive RealNVP and SESaMo can be applied. Indeed, the canonicalization approach could not straightforwardly be applied.

Fig. 13 demonstrates the greater capability of SESaMo to incorporate exact and broken continuous symmetries to enhance the model training and convergence. Moreover, this is further confirmed by the speed of convergence to a relatively high ESS as a function of training time, as shown in Fig. 14. After only seven minutes of training (one a single A100 NVIDIA GPU), the ESS achieved by SESaMo already surpasses 60% for both $\alpha = 0$ and $\alpha = 0.005$. In contrast, RealNVP, lacking the inductive bias induced by stochastic modulation, struggles to learn meaningful of the target density. The low ESS reflects this failure in learning the target probability density, as also shown in the RealNVP plots from Fig. 13.

## I  DETAILS OF NUMERICAL EXPERIMENTS

In this section, we present details of the numerical simulations and the hyperparameters used in the main paper. All NFs are trained on a single A100 NVIDIA GPU, using floating precision. For the Hubbard model, however, double precision is used to ensure numerically stable estimation of the fermion determinant. The Adam optimizer (77) is employed with a learning rate of $5 \times 10^{-4}$. Additionally, a learning rate scheduler is used: if the standard deviation of the loss has not changed over the last 2000 epochs, the learning rate is multiplied by a factor of 0.92. The learning rate is bounded from below at $1 \times 10^{-6}$. As discussed in Sec. 3.2, the gradients are estimated with the REINFORCE algorithm for all experiments with SESaMo. For the VMoNF method, the number of NFs is chosen accordingly to the number of symmetry sectors, i.e., four in the Hubbard case and eight for the Gaussian Mixture Model. For the $\phi^4$ theory we chose eight NFs as there are no clear symmetry sectors for the $U(1)$ symmetry. The remaining experiment-specific hyperparameters are summarized in Tab. 3. For FAB, the hyperparameters are chosen similarly, while for the Hubbard $18 \times 100$ system they are reduced to enable training on a single GPU. The number of training steps is chosen to match the training time of the other architectures.

---

[7]For VMoNF it was used Tanh, due to numerical instabilities.

[8]This variance was only used for the RealNVP and VMoNF models to alleviate mode-dropping.

## LARGE LANGUAGE MODEL USAGE

This work was partly supported by large language models. The usage contains polishing previously written text, discovering related work and assisting coding.

