# OpenReview forum: "SESaMo: Symmetry-Enforcing Stochastic Modulation for Normalizing Flows"
_ICLR.cc/2026/Conference — ICLR 2026 Poster_

### Official Review · Reviewer_Kqpf · 2025-10-28

**Soundness:** 4
**Presentation:** 3
**Contribution:** 3
**Rating:** 6
**Confidence:** 2

**Summary:**

The paper introduces a new way to incorporate symmetries into discrete normalizing flows, especially suited for symmetry breaking. In this scheme the NF always maps to the same symmetry sector (canonical mode or cell). Samples are then possibly transformed to one of the other symmetry sectors (picked stochastically according to learned probability). To allow gradient flow to the parametrization of this “stochastic modulation”, the authors introduce the “self-reparametrized KL divergence” (bijectivity penalty; REINFORCE estimator to differentiate through the random variable). Since the learned probabilities can differ between the modes, the method allows for symmetry breaking, i.e. different mass per mode. The authors test on a Gaussian mixture with exact and broken Z8 symmetry, 2D complex $\phi^4$ scalar field with U(1) symmetry, and Hubbard model with Z4 symmetry.

**Strengths:**

- The idea is technically sound and novel
- The paper shows clear benefits in terms of effective sample size (ESS) and lower KL on the tasks considered, especially for symmetry breaking

**Weaknesses:**

- The proposed stochastic modulation seems to only work for discrete symmetries
- The symmetry sectors (and the corresponding transformations from the canonical cell) must be known a priori

**Questions:**

- Appendix A and B (especially figure 5) are quite crucial to understanding the method. In my opinion they should be in the main text, possibly at the cost of moving parts of the background into the appendix
- How does accuracy/training complexity scale with dimension and number of symmetry sectors?
- Can stochastic modulation be used with continuous NF (like flow matching)?

---

> ### Author Response · Authors · 2025-11-21
>
> We thank the reviewer for their careful reading of our manuscript. We address their concerns and questions below.
>
> > W1: The proposed stochastic modulation seems to only work for discrete symmetries
>
> As  highlighted in the introduction our approach applies to *both* discrete and continuous symmetries and our work provides results for both cases in Table 1 in the main text. Furthermore, Appendix F explains in detail how the algorithm can be extended to continuous symmetry groups, and Appendix H.4 provides corresponding figures for the complex $\phi^4 $ theory, with both a broken and exact $U(1)$ symmetry. We acknowledge this aspect was not sufficiently emphasized  and we therefore added some clarifying sentence emphasizing this in the revised manuscript.
>
> > W2: The symmetry sectors (and the corresponding transformations from the canonical cell) must be known a priori
>
> We thank the reviewer for highlighting this aspect. Indeed, the symmetry sectors and corresponding transformations need to be known a priori, which is usually the case in physics and chemistry applications. Moreover, this limitation is in principle shared by all equivariant approaches. In contrast, methods that attempt to learn the symmetry sectors — such as VMoNF — clearly underperform in our experiments and fail to accurately reproduce the target distribution. We thus do not consider this a weakness of our approach.
>
>
> > Q1: Appendix A and B (especially figure 5) are quite crucial to understanding the method. In my opinion they should be in the main text, possibly at the cost of moving parts of the background into the appendix
>
> We thank the reviewer for this suggestion. We moved parts of Appendix A and Appendix B, including Figure 5, into the main manuscript—while keeping the detailed technical discussion in the appendix.
>
> > Q2: How does accuracy/training complexity scale with dimension and number of symmetry sectors?
>
> We thank the reviewer for raising this important question. The scaling of the approach with increasing system size depends strongly on the specific problem. For instance, in the 2D Hubbard model, the number of symmetry sectors grows with the system size $N_x$ as $2^{N_x-1}$. This in principle introduces an exponentially increasing number of breaking parameters. However, many of these sectors can be related through additional symmetries present in the system, such as spatial translation symmetry, which substantially reduces the number of *independent* breaking parameters. In our experiments, we have tested system sizes up to $N_x = 18$, and despite the large number of sectors, we find that the effective number of independent breaking parameters remains in the order of $\mathcal{O}(100)$. Crucially, we also do not observe any degradation in performance despite the larger amount of breaking parameters.
> To further assess the scalability and the computational costs, we conducted additional experiments on larger lattice volumes for the Hubbard model ($N_t \times N_x = 100 \times 18$).  These results indicate a quadratic scaling of computational cost with system size, which we consider mild and favorable.
>
> > Q3: Can stochastic modulation be used with continuous NF (like flow matching)?
>
> Yes, in principle, stochastic modulation can be combined with a wide range of models beyond coupling-based NFs, including continuous NFs, diffusion models, and flow matching. Exploring such extensions is an interesting direction for future work.

---

> > ### Comment · Reviewer_Kqpf · 2025-11-24
> >
> > Thank you for the clarifications regarding continuous symmetries.
> >
> > Where can I find the additional experiments on larger lattice volumes / system sizes for the Hubbard model? I think a table or plot that compares wall clock time and final accuracy metrics for different sizes would be helpful

---

> > > ### Author Response · Authors · 2025-11-28
> > >
> > > In Table 1 of the updated manuscript, we have added the new results for the ESS for $V = N_x \times N_t = 18\times 100$ for SESaMo and Canonicalization. The simulations for FAB and VMoNF are still running and we plan to include them in the camera-ready version of the paper. Additionally, we have added new methodological details on stochastic modulation for the Hubbard model with $N_x > 2$ in App. E.3. We believe that these additions further support SESaMo’s applicability to larger systems.
> > >
> > > In the table below, we provide additional results on the scaling with increasing $N_t$ and $N_x$. We report the wall-clock time required for training SESaMo until a 50\% acceptance rate is reached. We consider both cases of fixing $N_x = 2$ with $N_t = [20,30,40,50]$ and for $N_x = [2,4,8,12,18]$ with a fixed $N_t = 100$. As stated in a previous answer, we observe a quadratic increase in training time.
> > >
> > > Wall-clock time to reach 50\% acceptance for varying $N_t$ with fixed $N_x = 2$:
> > >
> > > | $N_t$ | Time (min)  |
> > > |-----|-------|
> > > | 20  | 0.05(2)  |
> > > | 30  | 0.08(2)  |
> > > | 40  | 0.13(2) |
> > > | 50  | 0.27(2) |
> > >
> > > Wall-clock time to reach 50\% acceptance for varying $N_x$ with fixed $N_t = 100$:
> > >
> > > | $N_x$| Time (min)  |
> > > |-----|-------|
> > > | 2  | 0.177(6)  |
> > > | 4  | 0.282(6)  |
> > > | 8  | 0.766(6) |
> > > | 12  | 1.340(7) |
> > > | 18  | 3.00(36) |

---

### Official Review · Reviewer_jKsj · 2025-10-31

**Soundness:** 3
**Presentation:** 3
**Contribution:** 3
**Rating:** 6
**Confidence:** 3

**Summary:**

This paper introduces SESaMo (Symmetry-Enforcing Stochastic Modulation), a framework that integrates group symmetry constraints into normalizing flows via a stochastic modulation mechanism. Specifically, the method augments a base flow g_\theta with a family of symmetry transformations S_u, where u is sampled from a learnable distribution p_{S,b}(u). This allows the model to both enforce exact symmetries and capture symmetry breaking. The approach is evaluated on several discrete-symmetry benchmarks, showing improved sampling performance and better mode coverage compared to RealNVP, canonicalization-based flows, and VMoNF.

**Strengths:**

The paper proposes a creative mechanism to integrate group symmetries into flow-based generative models using stochastic modulation. This differs from previous deterministic equivariant designs (e.g., Equivariant Flows) by introducing a learnable mixture structure that explicitly models symmetry breaking. Experiments demonstrate that SESaMo achieves competitive or superior ESS on discrete-symmetry datasets while maintaining interpretability of the learned symmetry-breaking parameter b.

**Weaknesses:**

1. The bijectivity penalty used to enforce separability is heuristic and not theoretically justified. The experimental comparisons only include RealNVP, VMoNF, and canonicalization methods. These are several years old and do not represent the current state of equivariant or symmetry-aware generative modeling. Modern approaches, including equivariant flow mathcing and equivariant diffusion models, should at least be discussed or justified as not directly applicable. Without this clarification, it is unclear whether SESaMo’s advantage is due to the stochastic modulation itself or simply the choice of older baselines.

2. The method assumes that each symmetry sector maps to a distinct, non-overlapping region of configuration space, which ensures bijectivity but restricts applicability to simple discrete symmetries.

3. The bijectivity penalty used to enforce separability is heuristic and not theoretically or experimentally justified.

4. The experiments focus on low-dimensional or lattice-based toy systems. There is no demonstration on realistic and high-dimensional systems, where the claimed symmetry advantages would be most meaningful.

**Questions:**

1. Could the authors clarify whether recent equivariant generative models cannot be directly applied to the symmetry-enforcing scenarios considered here? If so, please explain the technical limitations preventing their use. Otherwise, their inclusion as baselines would significantly strengthen the experimental evidence.

2. Could the authors discuss the potential of extending SESaMo to more complex or continuous symmetry groups?

3. The bijectivity penalty appears heuristic. How sensitive is the training process to the choice of its parameters (A and B)? Have the authors tried alternative formulations?

---

> ### Author Response · Authors · 2025-11-21
>
> > W1:The bijectivity penalty used to enforce separability is heuristic…
>
> We thank the reviewer for the valuable comments. We address the question on the penalty term in W3 and Q3 below. As for  the baselines, we have  included the [FAB 2023] baseline as suggested by reviewer HyQJ (see Q2 for HyQJ), commented on baselines for equivariant generative modeling above (see W2 for HyQJ), and discuss further baselines in our response to Q1 below.
>
>
> > W2: The method assumes that each symmetry sector…
>
>
> As stated in the manuscript (see Appendix E and F  in the original manuscript), our method can be applied to both discrete and continuous symmetries (exact and broken). We provide additional clarification in our response to Q2 below.
>
> Furthermore, because multiple reviewers asked about SESaMo’s capability to handle both continuous and discrete symmetries, we have added clarifying remarks to the revised manuscript, specifically in the main text.
>
>
> > W3: The bijectivity penalty used to enforce separability is heuristic…
>
>
> We thank the referee for this comment and respectfully disagree that the bijectivity penalty is simply heuristic. The term is well-justified theoretically, as it is used to constrain to the canonical cell which is  a priori identified by the symmetries of the system one wants to model. For more details on the parameters A and B, we refer to our reply Q3 below.
>
> > W4: The experiments focus on low-dimensional or lattice-based toy systems…
>
> As the first method capable of learning broken symmetries, our work provides a proof-of-concept demonstration on low-dimensional systems where the ground truth is analytically accessible. To further assess scalability, we have performed additional experiments on larger lattice volumes, specifically for the Hubbard model ($N_t \times N_x = 100 \times 18$). These results show a quadratic scaling of computational cost with system size, supporting the effectiveness and practicality of our symmetry implementation.
> Extensions to even larger and more realistic system sizes represent an important direction for future work.
>
>
> > Q1: Could the authors clarify whether recent equivariant generative models…
>
> We thank the reviewer for the question. Equivariance between prior and target requires both distributions to share the same symmetry. In our case, while we know the approximate symmetry of the target (e.g., the approximate $\mathbb{Z}_4$ symmetry in the 2D Hubbard model), its analytical form is unknown, so we cannot construct a prior with exactly matching symmetry. Thus, equivariant methods would face the same issue as canonicalization in our manuscript: as symmetry breaking increases, their performance deteriorates. Our method avoids this issue by explicitly employing a non-equivariant transformation.
>
> Moreover, equivariant flow-matching and diffusion models require samples from the target distribution, but we work with Boltzmann distributions that are hard to sample using standard MCMC (such as the Hubbard model). Our data-free training setup therefore makes these methods inapplicable.
>
> > Q2: Could the authors discuss the potential of extending SESaMo to more complex or continuous symmetry groups?
>
> In Appendix F we discuss how SESaMo can be extended to continuous symmetry groups. Moreover, in Table 1, we provide corresponding results for the complex $\phi^4$ theory, with both broken and exact U(1) symmetry (i.e., continuous symmetry group). We acknowledge that this was not sufficiently emphasized in the main text and we added a clarifying sentence in the revised manuscript.
>
> > Q3: The bijectivity penalty appears heuristic…
>
> We thank the reviewer for raising this point. As mentioned above, the penalty term is theoretically well-motivated, as it allows to restrict to the canonical cell and is therefore determined *a priori* by the symmetries of the underlying model.
>
> However, an infinite-amplitude Heaviside step function would not yield the required gradient for the samples to be pushed into the canonical cell. Therefore one has some flexibility in choosing its smooth approximation (e.g., tanh or sigmoid) and thus corresponding parameters A and B.
>
> Nonetheless, as long as the gradient remains sufficiently moderate near the boundary and the amplitude is sufficiently large, different functional choices lead to the same behavior. To support this, we have repeated the experiments with alternative values of A and B, which yield comparable performance if A > 10 and B < 1000. For smaller A, the penalty is too weak to contribute meaningfully to the loss, while for larger B, the gradient of the sigmoid vanishes outside the boundary region. We provide values for the ESS of different A and B in the table below which are gained when training SESaMo for the Hubbard model.
> |A=1000, B=|1|10|100|1000|10000|
> |-|-|-|-|-|-|
> |ESS|0.9983|0.9983|0.9987|0.9987|0.3212|
>
> |B = 100, A=|1|10|100|1000|10000|
> |-|-|-|-|-|-|
> |ESS|0.2179|0.9988|0.9988|0.9987|0.9985|

---

> > ### Comment · Reviewer_jKsj · 2025-11-27
> >
> > Thank authors for the rebuttal and for the additional clarifications, especially the discussion and modifications concerning the continuous setting.
> >
> > I have one remaining question regarding the new experiments mentioned in the response. The authors stated: “we have performed additional experiments on larger lattice volumes, specifically for the Hubbard model.” I would like to confirm whether these results correspond to the Hubbard model experiments already shown in the current manuscript. I don't find information about the lattice size in the paper’s experimental section.

---

> > > ### Author Response · Authors · 2025-11-28
> > >
> > > We thank the reviewer for raising this point. As reviewer Kqpf asked the same question, we kindly refer to our response below.

---

### Official Review · Reviewer_goWZ · 2025-11-01

**Soundness:** 3
**Presentation:** 2
**Contribution:** 2
**Rating:** 4
**Confidence:** 3

**Summary:**

This paper introduces SESaMo (Symmetry-Enforcing Stochastic Modulation), a simple and general framework for incorporating both exact and broken symmetries into normalizing flows (NFs). The method augments a standard flow with a symmetry transformation whose selection probabilities are learnable, allowing the model to capture multiple symmetric modes and symmetry-breaking effects without having to enforce symmetries as hard constraints within the architecture. Experiments on Gaussian mixtures and lattice field theory benchmarks ($\phi^4$ and Hubbard models) demonstrate improved sample efficiency and effective sample size compared to normalizing flow baselines, including with canonicalization.

**Strengths:**

* The proposed method is a simple and flexible way to impose both exact and broken symmetries on NFs without requiring explicitly equivariant architectures or canonicalization.
* The work is technically sound and supported with solid empirical validation.
* Furthermore, the proposed method integrates naturally with standard NFs (e.g. RealNVP) and requires relatively minimal overhead (a REINFORCE estimator to enable gradients through a stochastic variable).

**Weaknesses:**

* While the proposed method is elegant in principle, the exposition is overly abstract which makes the main idea underlying the method difficult to understand. For example, the authors provide several intuitive examples in increasing generality, but relegate them to the appendix (E, F). The paper could benefit from a clear algorithm box/pseudocode implementation describing the key components (flow, modulation map, and probability weights) and, in increasing generality, how each can be set or augmented with learned components. I believe this would make the work much more accessible.
* The paper presents the method and empirical results, but could benefit from a formal analysis of conditions for convergence, expressivity, or relations to existing equivariant NFs. For example, as far as I understand, the REINFORCE estimator and stochastic variable introduced for the modulation map is likely to introduce additional variance, but it is unclear to what extent this affects the training stability or the consistency of the results (see questions.) Additional experiments that quantify the variability of the results with SESaMo may further strengthen the paper.

**Questions:**

* If I understand correctly, the stochastic variable + REINFORCE estimator used in the modulation map is likely to increase variance. Furthermore, since the approach still minimizes reverse KL, the approach remains mode-seeking and it seems possible (with poor initialization or high-variance training) that a mode may be entirely missed. Did the authors observe this potential issue?
* Did the authors evaluate the method on datasets beyond toy or physical models where symmetry-enforcement is important, e.g. rotation/reflections with natural images?

---

> ### Author Response · Authors · 2025-11-21
>
> We thank the reviewer for their careful reading of our manuscript. We refer them to the general reply for a comprehensive discussion of the broader points raised by multiple reviewers. Below, we address the reviewer’s specific concerns and questions in detail.
>
> > W1: While the proposed method is elegant in principle…
>
> We thank the reviewer for their suggestion. We added a pseudocode to the revised manuscript. In addition, in the revised version of the manuscript we bring key aspects from Appendix E and Appendix F into the main text, see for example Eq. (38). As suggested by the reviewer this should make the work overall more accessible, without altering the flow of the paper.
>
> > W2: The paper presents the method and empirical results…
>
> We thank the reviewer for the thoughtful question. The key conditions for our approach to work are as follows: prior determination of the symmetry sectors, a well-defined canonical cell, a target density with zero probability at the border of the canonical cell, and a sufficiently expressive neural network architecture (see Table 3).
> To ensure the fairest comparison with respect to Equivariant NFs we compared our method to symmetry-incorporating flows (equivariant NFs) used in physics and chemistry. Those are equivariant flows based on canonicalization (see e.g. [Boyda et al., PRD (2021)](https://journals.aps.org/prd/abstract/10.1103/PhysRevD.103.074504).
> Other methods, such as diffusion models, differ fundamentally as existing approaches would require a large amount of data from the target and are therefore not directly applicable to our setting.
> We agree that, in principle, the stochastic variable could introduce additional variance during training. However, in Fig. 9 and Fig. 14 we analyze training time and convergence over ten independent initializations. Furthermore, note that the variance for SESaMo is smaller compared to the other baselines. This is also reflected in Table 1, where the uncertainty is the standard error of the mean, and SESaMo achieves at least two orders of magnitude smaller uncertainty compared to the baselines.
>
> > Q1: If I understand correctly, the stochastic variable + REINFORCE estimator…
>
> Regarding the variance, we refer to  our reply above.
> Regarding the mode-seeking behavior of the reverse KL, we note that a key motivation for incorporating symmetries into a normalizing flow is that one reduces to learn only a single mode (the canonical cell), in order to avoid mode-dropping. However, if a canonical cell were to contain multiple modes that are not related by another symmetry (case for which we do not have an example at hand yet), then, in principle, SESaMo could be mode dropping..
>
> > Q2: Did the authors evaluate the method on datasets beyond toy or physical models where symmetry-enforcement is important, e.g. rotation/reflections with natural images?
>
> The main motivation for SESaMo is to provide an alternative approach to established Monte Carlo (MC) simulations in physics and chemistry. MC approaches often suffer from ergodicity issues when the target probability density is multimodal. Furthermore, in physics we normally do not have access to target data which we could use to train a generative model (i.e., similar to computer vision use cases). For this reason we focus exclusively on training in the data-free setting (toy data and real physics use cases) while considering the application of SESaMo in e.g., a computer vision context (where target data are required) to be out of scope for this work.

---

### Official Review · Reviewer_HyQJ · 2025-11-02

**Soundness:** 2
**Presentation:** 3
**Contribution:** 2
**Rating:** 2
**Confidence:** 4

**Summary:**

The paper presents a sampling method that handles symmetries in unnormalized distributions in a principled way. It maps the random tensor of interest $x$ to a reduced space $\tilde x$. It learns a Normalizing Flow model to sample $\tilde x$ and then map it, along with a latent variable $u$, back to $x$.

Experiments are conducted on several distributions to validate the effectiveness of the proposed method.

**Strengths:**

- The paper introduces a systematic way to incorporate symmetries in NF.
- Experimental testbed is good. The complex $\phi^4$ and Hubbard models are known to be challenging distributions.
- The idea of using a penalty term to enforce bijectivity is interesting.
- The paper is well written and is mostly comprehensible.

**Weaknesses:**

I think several essential questions/issues are overlooked both in theory and experiments.
- Experimental results report only ESS and RKL (reverse KL divergence). However, the performance on two metrics can be good (i.e., high ESS and low RKL) even in the case of mode collapse (see [AdvNF 2025]), i.e., when the model learns and explores only a part of the target distribution. A standard metric robust to this problem is the negative log-likelihood (NLL).
- The baselines are weak. The state of the art baselines such as [FAB 2023] and [IDEM 2024] should be used.
- Are all $S_{T,u}$ independently trained for each $u$? If no, this could be a problem because the symmetry breaking bias may cause a probability mass drift even within a mode (of symmetry). If yes, the number of parameters involved will shoot up, and this should be discussed in the paper (comparing model sizes and training times of different methods/models). Or is the model conditioned on $u$ (i.e., $u$ is an input)?

In the core, the proposed approach factorizes $q(x)$ as $\sum_u q(x|u)p(u)$, where $u$ is a random variable corresponding to a symmetry mode. Theoretically, the contribution seems modest. More rigorous experiments could make the paper more appealing.

The presentation could be simplified to highlight the core contribution. There are too many variables and equations, and many are superfluous.


[AdvNF 2024] AdvNF: Reducing mode collapse in conditional normalising flows using adversarial learning, SciPost Phy. 2024
[FAB 2023] FLOW ANNEALED IMPORTANCE SAMPLING BOOTSTRAP, ICLR 2023
[IDEM 2024] Iterated Denoising Energy Matching for Sampling from Boltzmann Densities, NeurIPS 2024

**Questions:**

- Please answer my concerns raised in the Weaknesses section.
- It seems the paper considers only discrete symmetries as $u$ is discrete, and not continuous ones. But the experiments are performed for U(1) symmetry too. What is $u$ in that case?
- Although bijectivity has been enforced by a penalty term, its effectiveness has not been assessed. Experiments could be added to address this.
- Why do we need to learn $C_{T,z}$? Can't it be a heuristic map, or can't we sample $\tilde z$ directly? I would appreciate it if another example(s) could help me understand better.

- Minor typos:
  - Line 228: It think it should be $g_\theta(z)$ instead of just $g_\theta$.
  - Line 283: a -> an

---

> ### Author Response · Authors · 2025-11-21
>
> We thank the reviewer for reading our manuscript. In our answers below, we specifically address the reviewers questions and concerns.
> > Q1: Experimental results report only ESS and RKL (reverse KL divergence). However, the performance on two metrics can be good (i.e., high ESS and low RKL) even in the case of mode collapse (see [AdvNF 2025]), i.e., when the model learns and explores only a part of the target distribution. A standard metric robust to this problem is the negative log-likelihood (NLL).
>
> We thank the reviewer for the suggestion. As emphasized in [AdvNF 2025], “NLL estimates how efficiently the model fits the data”, however, in our work we fully rely on the reverse KL and train without any data. Nonetheless, to account for mode collapse we added the empirical (visual) validation that our method has no mode collapse in the result section.
>
> Moreover, while the ESS does not capture mode collapse, the reverse KL (with including normalizing factor $\log Z$) does show larger values if a mode is missing. The KL divergence that we show in Table 1 includes the normalization factor and only converges to zero if there is no mode collapse, which is the case for our SESaMo results.
>
> > Q2: The baselines are weak. The state of the art baselines such as [FAB 2023] and [IDEM 2024] should be used.
>
> We thank the reviewer for pointing to other baselines. We implemented the method from [FAB 2023] and report the evaluated metrics here. FAB is very robust to mode dropping arising from the mode-covering loss function, which makes this method a very solid baseline. However, using the symmetry knowledge, we are still able to outperform it. We will include this benchmark in the table of the final manuscript.
>
>  |       | GMM | broken GMM | $\phi^4$ theory | broken $\phi^4$ theory | Hubbard model |
> |-|-|- |-| -|-|
> | ESS |  0.81  |  0.79   |    0.063     |   0.055      | 0.97   |
> | KL | 0.86  |   0.86   |      N/A      |    N/A       | 0.25  |
>
>
> > Q3: Are all $S_{T,u}$ independently trained for each $u$? …
>
> We thank the reviewer for this question. We believe there might be a misunderstanding.The map $S_{T,u}$ itself is not trained. Depending on the value of $b$, we sample $u$, which then determines $S_{T,u}$. The number of parameters $b$ is fixed a priori and is dictated by the symmetry of the system under consideration. For the exact GMM, we have 0 parameters for the stochastic modulation since all modes have equal weight; for the symmetry-broken GMM, we have 7 parameters; for the Hubbard model, we have 1 parameter because there are only two modes with different weights. The continuous symmetry breaking in the $\phi^4$ theory is parametrized by a spline flow which learns the rotation angle (see App. F in the manuscript). As the complexity of the broken symmetry increases, the number of parameters $b$ increases accordingly. We will add a clarifying comment on this point to the manuscript.
>
>
>
> > Q4: It seems the paper considers only discrete symmetries as $u$ is discrete, and not continuous ones. But the experiments are performed for U(1) symmetry too. What is $u$  in that case?
>
> In Appendix F we explain how the algorithm can be extended to continuous symmetries.We also refer to Tab. 1 and the corresponding figures in App. H.4. Specifically, for U(1), we sample $u$ uniformly from $[0,1)$ and map it to the circle via the rotation $R_{h_b(u)}$, where $h_b(u)$ is the rotation angle (see Eqs. (41) and (43)). We thank the reviewer for the question as it suggests that this point was not sufficiently clear in the original version of the manuscript. We will revise the text accordingly to improve clarity.
>
> > Q5: Although bijectivity has been enforced by a penalty term, its effectiveness has not been assessed. Experiments could be added to address this.
>
> We thank the reviewer for their suggestion. To evaluate the impact of the penalty term, we measured the average bijectivity violation across all experiments presented in the paper. In all cases, fewer than 10^{-6} of the samples lay outside the canonical cell, thus violating bijectivity. This demonstrates that the penalty term results in a negligible violation of bijectivity.
>
>
>
> > Q6: Why do we need to learn $C_{T,z}$? Can't it be a heuristic map, or can't we sample \tilde{z} directly? I would appreciate it if another example(s) could help me understand better.
>
>
> We again believe there might be a slight misunderstanding.The map $C_{T,z}$ is defined a priori and is *not* learned in the training process. Furthermore, as mentioned in Sec. 2.3.2 we stress that the transformation $C_{T,z}$  is not needed for SESaMo, but only required by the standard canonicalization approaches.

---

### Author Response · Authors · 2025-11-21
**General statement about continuous symmetries and data free training**

We thank all reviewers for their thoughtful and constructive feedback. Several comments raised overlapping questions, which we address here in a general response. We provide a first draft of the updated paper where changes are highlighted in blue.

**Continuous Symmetries**:

First, while the main text focuses on discrete symmetry groups, our method is **not limited to discrete symmetries**. We explicitly apply the algorithm to the complex $\phi^4$ theory, which features a broken U(1) continuous symmetry. As mentioned in the introduction, our approach applies to both **continuous and discrete symmetries**. In Table 1 in the main text of the paper we provide results for both. Furthermore, Appendix F provides the technical details on how to extend SESaMo to continuous groups. Finally, Appendix H.4 presents the corresponding results. We acknowledge that these important appendices were not sufficiently emphasized in the main manuscript and we highlighted them more clearly in the revised version.

**Data Free training**:

SESaMo is intended for sampling from challenging Boltzmann distributions in settings where *in principle* no target samples are accessible, placing it within the class of so-called Boltzmann Generators. In many physics applications, conventional Monte Carlo techniques are theoretically able of drawing samples; however, they frequently encounter severe ergodicity breakdowns, topological freezing, or long autocorrelation times, leading to sampled configurations that do not represent the desired distribution *in practice*. This motivates our data-free training strategy. Accordingly, comparisons with methods that **require** samples from the target distribution are not meaningful in such regimes.
We hope that these clarifications fully address the shared concerns.

---

### Meta-Review · Area_Chair_Nkyj · 2026-01-05

**Summary:**

Summary of reviewers' concerns:

1. Is the bijectivity requirement fulfilled? (HyQJ)
2. Can symmetries be continuous? (HyQJ, jKsj, Kqpf)
3. Is mode collapse observed? (HyQJ, goWZ)
4. What about FAB and IDEM baselines? (HyQJ)
5. Scaling to larger problems and to other settings (e.g. forward KL)? (jKsj)

**Reviewer Concerns:**

They were largely addressed:

1. Empirically, this is fine.
2. Continuous symmetries are supported.
3. Mode collapse might still be a problem within each cell, but the method helps mitigating mode collapse between cells.
4. FAB was implemented and is outperformed. The authors seem to have missed IDEM, but since it is a diffusion model and somewhat orthogonal, the data point was not important for deciding on acceptance/rejection.
5. They show promising performance on a larger Hubbard system.

I stongly encourage the authors to include recent diffusion/flow matching baselines for comparison.

**Reviewer Scores:**

I think that all reviewers' main concerns were addressed. jKsj and Kqpf already recommended acceptance before the rebuttal. HyQJ's questions were largely addressed with clarifications as well as an additional baseline and experimental details. goWZ might have increased their score since the mode collapse was clarified.

I strongly encourage the authors to improve the presentation. Several reviewers highlighted room for improvement, and a good paper needs to be accessible so that it is adopted.

---

### Decision · Program_Chairs · 2026-01-26

Accept (Poster)